# Artificial intelligence enables precision diagnosis of cervical cytology grades and cervical cancer

Jue Wang[1,2,12], Yunfang Yu[1,3,4,5,12], Yujie Tan[1,3,4,12], Huan Wan[1,2,12], Nafen Zheng[1,2,12], Zifan He[1,3,4], Luhui Mao[1,3,4], Wei Ren[1,3,4], Kai Chen[1,3,4], Zhen Lin[6], Gui He[1,2], Yongjian Chen[7], Ruichao Chen[8], Hui Xu[9], Kai Liu[6], Qinyue Yao[6], Sha Fu[1,2], Yang Song[1,2], Qingyu Chen[10], Lina Zuo[10], Liya Wei[10], Jin Wang[6] ✉, Nengtai Ouyang[1,2] ✉ & Herui Yao[1,3,4,11] ✉

Cervical cancer is a significant global health issue, its prevalence and prognosis highlighting the importance of early screening for effective prevention. This research aimed to create and validate an artificial intelligence cervical cancer screening (AICCS) system for grading cervical cytology. The AICCS system was trained and validated using various datasets, including retrospective, prospective, and randomized observational trial data, involving a total of 16,056 participants. It utilized two artificial intelligence (AI) models: one for detecting cells at the patch-level and another for classifying whole-slide image (WSIs). The AICCS consistently showed high accuracy in predicting cytology grades across different datasets. In the prospective assessment, it achieved an area under curve (AUC) of 0.947, a sensitivity of 0.946, a specificity of 0.890, and an accuracy of 0.892. Remarkably, the randomized observational trial revealed that the AICCS-assisted cytopathologists had a significantly higher AUC, specificity, and accuracy than cytopathologists alone, with a notable 13.3% enhancement in sensitivity. Thus, AICCS holds promise as an additional tool for accurate and efficient cervical cancer screening.

Cervical cancer ranks fourth globally among the most common cancers and the fourth leading cause of cancer-related deaths[1–3]. Early screening plays a vital role in its effective prevention. Timely detection and intervention to halt the progression of precancerous cervical lesions are essential. However, there's a pressing need for accurate

screening platforms for early cervical cancer detection. Presently, screening methods primarily include cervical cytology, HPV testing, and DNA ploidy testing[4]. Cervical cytology screening known for its simplicity and cost-effectiveness, is recommended for population-based screening[5]. Nevertheless, there is a significant shortage of

[1]Guangdong Provincial Key Laboratory of Malignant Tumor Epigenetics and Gene Regulation, Sun Yat-sen Memorial Hospital, Sun Yat-sen University, Guangzhou, China. [2]Department of Cellular and Molecular Diagnostics Center, Sun Yat-sen Memorial Hospital, Sun Yat-sen University, Guangzhou, China. [3]Department of Medical Oncology, Sun Yat-sen Memorial Hospital, Sun Yat-sen University, Guangzhou, China. [4]Phase I Clinical Trial Centre, Sun Yat-sen Memorial Hospital, Sun Yat-sen University, Guangzhou, China. [5]Faculty of Medicine, Macau University of Science and Technology, Taipa, Macao, China. [6]Cells Vision (Guangzhou) Medical Technology Inc., Guangzhou, China. [7]Dermatology and Venereology Division, Department of Medicine Solna, Center for Molecular Medicine, Karolinska Institutet, Stockholm, Sweden. [8]Department of Pathology, The Third Affiliated Hospital of Guangzhou Medical University, Guangzhou, China. [9]Department of Pathology, Guangzhou Women and Children's Medical Center, Guangzhou Medical University, Guangzhou, China. [10]Department of Health Center, Sun Yat-sen Memorial Hospital, Sun Yat-sen University, Guangzhou, China. [11]Breast Tumor Centre, Sun Yat-sen Memorial Hospital, Sun Yat-sen University, Guangzhou, China. [12]These authors contributed equally: Jue Wang, Yunfang Yu, Yujie Tan, Huan Wan, Nafen Zheng. ✉e-mail: wangjin@cellsvision.com; ouynt@mail.sysu.edu.cn; yaoherui@mail.sysu.edu.cn

cytopathologists worldwide has led to over a 10% false negative rate in routine diagnosis[6]. In China the overall cervical cancer screening rate remains low compared with developed countries, largely due to its shortage of pathologists. As of 2022, China had only 20,400 registered pathologists, while the actual demand for them was nearly 100,000[7].

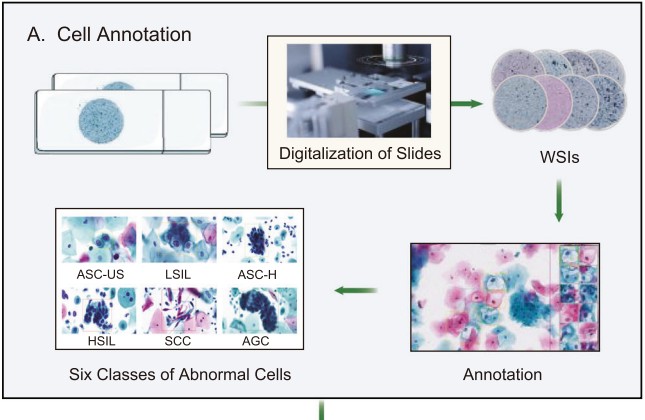

**A. Cell Annotation**

Digitalization of Slides

WSIs

ASC-US    LSIL    ASC-H

HSIL    SCC    AGC

Six Classes of Abnormal Cells

Annotation

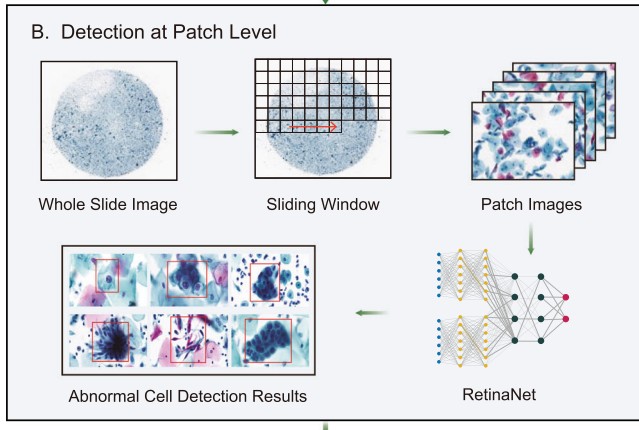

**B. Detection at Patch Level**

Whole Slide Image

Sliding Window

Patch Images

Abnormal Cell Detection Results

RetinaNet

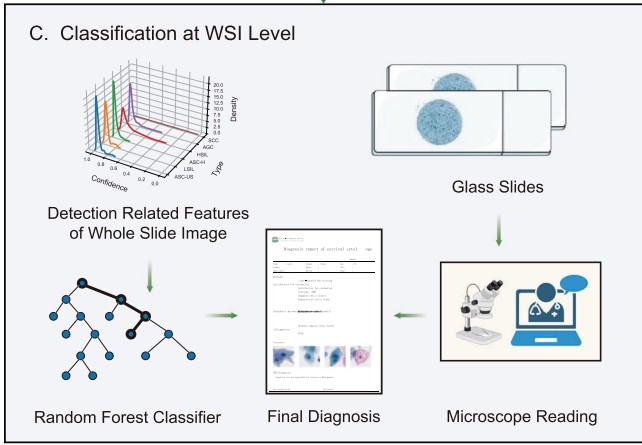

**C. Classification at WSI Level**

Detection Related Features of Whole Slide Image

Glass Slides

Random Forest Classifier

Final Diagnosis

Microscope Reading

**Fig. 1 | The AICCS workflow.** The acquisition and preprocessing of WSIs involve first digitalizing cervical liquid-based preparation samples collected and maintained by the sedimentation liquid-based preparation method (**A**). WSIs passing quality control undergo patch-level detection which involves dividing a WSI into smaller patches using a sliding window approach and annotating annotate abnormal cells based on the criteria defined by TBS 2014 (**B**). The output of the cell detection model serves as input for the WSI-level classification model. The WSI-level classification model utilizes the results from the patch-level cell detection model to generate possible cytology grades according to TBS 2014: ASC-US, LSIL, ASC-H, HSIL, SCC, and AGC (**C**). The content of microscope reading in Section C classification at WSI level are origin from BioRender. The 'Confirmation of Publication and Licensing Rights' is provided.

Hence, there's an urgent need to develop auxiliary tools for cervical cancer screening.

Artificial intelligence (AI) has swiftly advanced and reshaped various aspects of daily life. Innovations in methods like deep learning, encompassing convolutional neural networks, object detection models, ensemble learning approaches, and generative adversarial networks, have fueled the expansion of AI-driven applications and research in health-care. These breakthroughs hold promise for cost reduction and enhanced diagnoses[8–11]. Computer vision has found extensive use in disease detection. For example, in 2016, Google developed a model capable of detecting diabetic retinopathy with accuracy comparable to trained medical professionals[12]. Moreover, in 2019, they devised a model that surpassed doctors in identifying lung cancer[13]. Encouraging outcomes have also emerged in breast cancer detection, with numerous studies employing deep learning techniques for mammography and digital breast tomosynthesis classification[14–17]. These techniques have achieved excellent performance when assessed on extensive datasets.

The emergence of an AI-powered cervical cancer screening system holds promise for transforming cervical cancer diagnosis. Traditional grading methods in cervical cytology grading hinge on manual assessment by pathologists, which is time-consuming, subjective, and susceptible to inter-observer differences. While cervical cytology remains widely used, it heavily relies on the expertise and experience of cytopathologists. Microscope readings, for instance, suffer from poor reproducibility and susceptibility to various interfering factors. In contrast, AI screening offers potential advantages such as enhancing the consistency of cytopathological results, improving alignment with biopsy outcomes, boosting sensitivity, and reducing the risk of misdiagnosis[18,19]. In a prior study, researchers developed a comprehensive cervical liquid-based cytology smear AI-assistive TBS (AIATBS) diagnostic system[20]. This system utilized YOLOv3 for target detection, Xception, and patch-based models for target classification, and U-net for nucleus segmentation. The AI system was trained to accommodate 24 different classifications. However, its features presented challenges for applying it broadly in practice, as it actually increases the complexity of screening tasks. Rahaman et al. employed a deep learning-based hybrid deep feature fusion (HDFF) technique for the classification of cervical cytology[21], but it focused solely on the classification of squamous epithelial cells.

The objective of this study was to create an artificial intelligence cervical cancer screening (AICCS) system to aid in diagnosing cervical cytology grades and cervical cancer by analyzing whole-slide images (WSIs) of cervical cytology. The system underwent validation using multicenter, retrospective, and prospective population-based datasets, along with a randomized observational trial.

## Results

### Proposed an AICCS Model

In response to the growing need for precision medicine, deep learning has emerged as the preferred approach for automated intelligent medical diagnosis. To advance early cervical cancer screening, particularly in detecting abnormal or malignant cells in a cervical thin liquid-based test, we developed an Artificial Intelligence Cervical Cancer Screening (AICCS) system. In the conventional diagnostic process, after collecting and preparing smear samples onto a liquid-based slide, cytopathologists manually inspect them under a microscope to identify potential abnormal and malignant cells, which is time-consuming, draining for cytopathologists, and increasing the risk of misdiagnosis. Leveraging the latest Artificial Intelligence (AI) technology and the expertise of experienced cytopathologists, the AICCS was crafted with a deep-learning neural network and trained it with well-annotated WSIs. Figure 1 shows the workflow of cervical cancer diagnosis, which consists of three major stages: cell annotation (Fig. 1A), detection at the patch-level (Fig. 1B), and classification at the

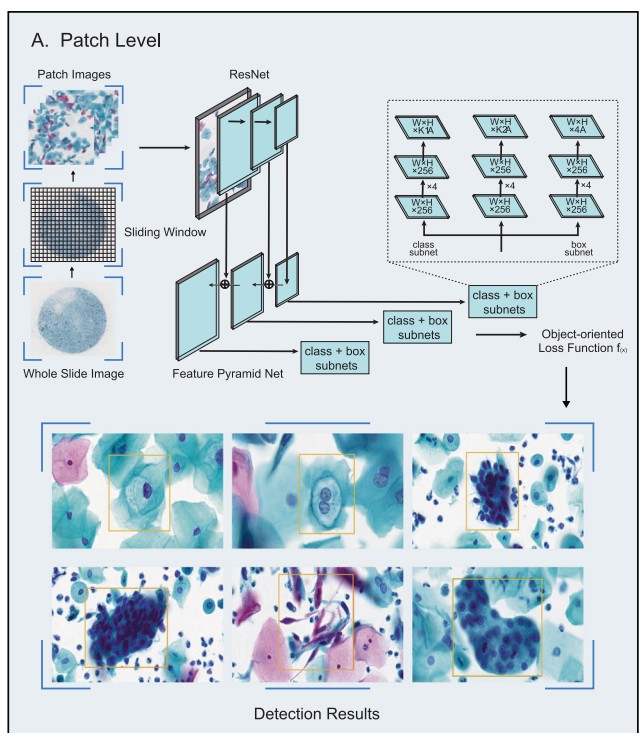

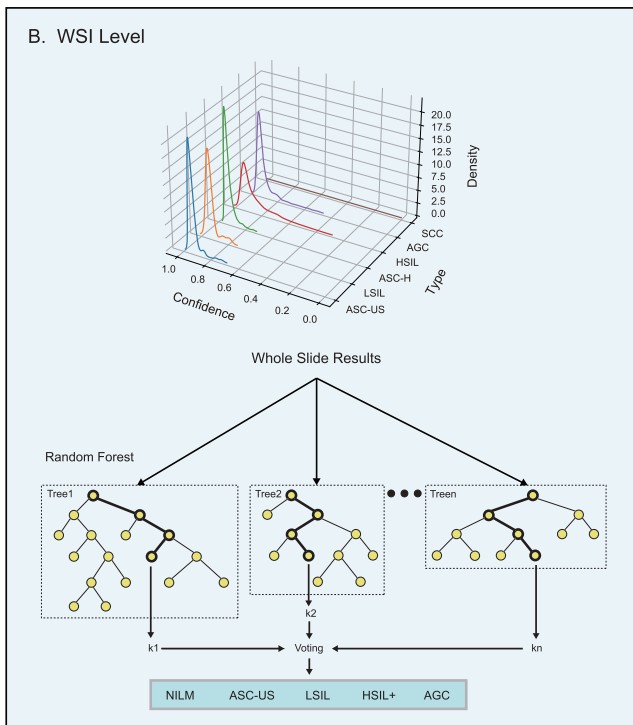

**Fig. 2 | The AICCS system algorithm.** We adopted RetinaNet, a one-stage object detection approach, as our anomaly cell detection model at the patch-level, and designed it to distinguish between six classes of abnormal cells: ASC-US, LSIL, ASC-H, HSIL, SCC, and AGC (**A**). The development of a WSI classification model involves generating features that encapsulate the statistical data derived from the patch-level detection model, which are then utilized to train the WSI classification model via the implementation of a random forest algorithm (**B**).

WSI-level (Fig. 1C). Annotation occurs solely during the training phase of the AICCS. During the operating process with the trained model, only digitization needs to be performed to obtain the whole-slide images (WSIs). Figure 1B shows a detection process that attempts to simulate the process of manual identification using a deep-learning object detection neural network to go through every small sliding window, or through patches of the WSIs. The third stage (Fig. 1C) involves aggregating all detections in stage two and performing classification at the WSI-level to provide a suggestion to cytopathologists. The last step is reviewing the AI's suggestion and verifying it with a WSI browser or the direct use of a microscope.

The AICCS, it is a combination of a deep-learning neural network and classical machine-learning algorithms. The AICCS algorithm comprises two major functional models: a patch-level cell detection model (Fig. 2A) and a WSI-level classification model (Fig. 2B). For model selection at the patch-level, the Retina and Faster-R-CNN algorithms exhibited similar performances (Supplementary Table 1). The AICCS's full computational model was further optimized and generated by comparing four algorithm combinations, namely, two patch-level detection models (Retina and Faster-R-CNN) with two WSI level classification models (random forest and DNN). Among them, the Retina-Resnet18-random forest algorithm was selected for its high performance in diagnosing cervical cytology grades. It achieved an AUC of 0.922 (95% CI 0.904–0.940) and a sensitivity of 0.906 (95% CI 0.875–0.932) (Supplementary Table 2). Thus, RetinaNet was selected as the abnormal cell detection model at the patch-level. Abbreviations are defined in Supplementary Table 3.

### Dataset characteristics for AICCS system development

To develop the AICCS system, we recruited a total of 16,056 eligible participants were recruited from three institutions serving as the training, validation, prospective, and randomized observational trial datasets (Supplementary Fig. 1). Among them, 11,468 participants from Sun Yat-sen University Sun Yat-sen Memorial Hospital (SYSMH) met our inclusion criteria for the model development, of these, 9316 were utilized for training the model, and 2152 were allocated for internal validation. Two external validation datasets were obtained from the Guangzhou Women and Children Medical Center (GWCMC) and the Third Affiliated Hospital of Guangzhou Medical University (TAHGMU), each comprising 600 participants. Additionally, the SYSMH prospective dataset comprising 2780 eligible participants. Lastly, 608 eligible participants from SYSMH were enrolled in the randomized observational trial (Supplementary Fig. 2).

To enhance model training, we increased the number of intraepithelial lesions cases in the training, internal validation, and external validation dataset, resulting in a higher proportion of positive cases compared to the epidemiological ratio. In the SYSMH training dataset and internal validation dataset, intraepithelial lesions accounted for 27.6% and 20.3%, respectively (Fig. 3A, B). In the external validation datasets from GWCMC and TAHGMU, the proportions of intraepithelial lesions were 44.7% and 32.7%, respectively (Fig. 3C, D) (Supplementary Table 4). Although the AICCS system was trained on a retrospective dataset, to verify the effectiveness in a realistic setting, prospective validation dataset and randomized observational trial datasets were constructed. In these datasets, the proportions of intraepithelial lesions were 4% and 7.7%, respectively, aligning with clinical practice where the positive rate range from 3 to 8% (Fig. 3E–H). The classification and proportion of cervical cytology for each dataset group are detailed in Supplementary Tables S4 and S5.

### Performance evaluation of the AICCS system on the validation datasets

Supplementary Table 6 presents the experimental results of AICCS across all cytological grades and other subgroups. It demonstrates that the AICCS system achieved high performance in identifying abnormal

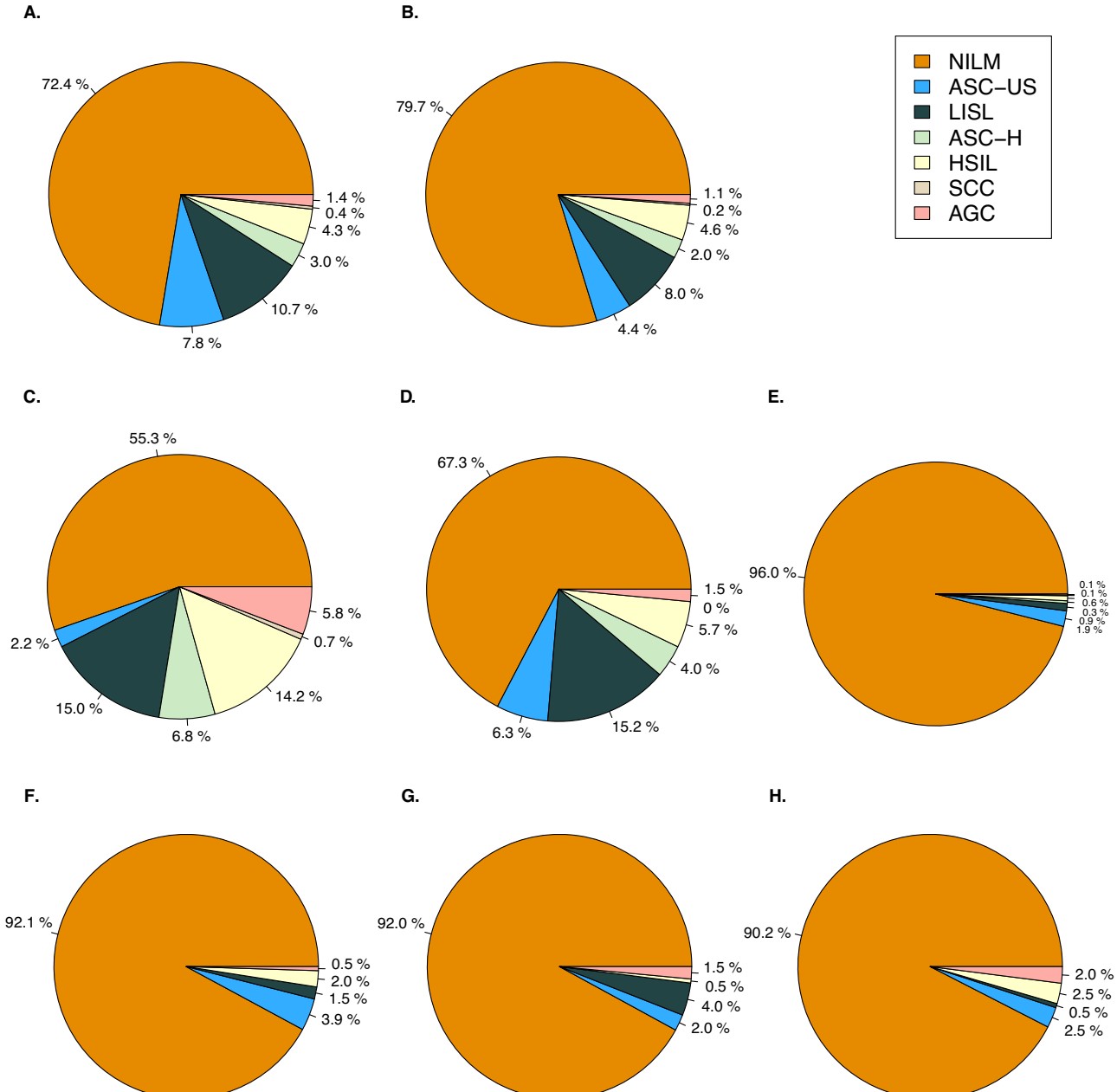

**Fig. 3 | Distribution of the cervical cytology grade in the training, validation, and randomized observational trial datasets.** **A** illustrates that the training dataset comprised 730 (7.8%) cases of ASC-US, 995 (10.7%) cases of LSIL, 279 (3.0%) cases of ASC-H, 401 (4.3%) cases of HSIL, 35 (0.4%) cases of SCC, and 131 (1.4%) cases of AGC. **B**–**D** depict that 20.3%, 44.7%, and 32.7% of cases had intraepithelial lesions in the internal validation dataset, GWCMC external validation dataset, and TAHGMU dataset, respectively. **E** shows that 4% of participants had intraepithelial lesions in the SYSMH prospective validation dataset. Detailed proportions of intraepithelial lesions in AICCS alone, cytopathologists, and AICCS-assisted cytopathologists in the randomized observational trial are shown in (**F**–**H**). Source data are provided as a Source Data file.

cell grades in both internal and two external validation datasets. Specifically, the AICCS system maintained sensitivity, accuracy, specificity, and AUC values above 0.800 in all internal and external validation datasets.

In conventional medical practice, subsequent management protocols vary based on the classification of cervical cytology grades. For cases classified as negative for intraepithelial lesion or malignancy (NILM), no additional investigative procedures are warranted. However, for cytological results categorized as low-grade squamous intraepithelial lesion (LSIL) or higher, colposcopy is recommended. In instances where atypical squamous cells of undetermined significance (ASC-US) are detected, human papillomavirus (HPV) testing is advised.

A negative HPV test suggests a follow-up cervical cytology screening after one year. while a positive result necessitates colposcopy. Based on these stratified protocols, subgroup analyses were conducted to assess the efficacy and outcomes of the AICCS system.

For subgroup analysis, the ASC-US+ category included ASC-US, LSIL, atypical squamous cells - cannot exclude HSIL (ASC-H), high-grade squamous intraepithelial lesions (HSIL), and squamous cell carcinoma (SCC). LSIL+ encompassed LSIL, ASC-H, HSIL, and SCC, while HSIL+ included ASC-H, HSIL, and SCC. As shown in Fig. 4, all subgroups of ASC-US+, LSIL+, and HSIL+ exhibited high AUC values. Furthermore, the AICCS system achieved higher AUC values with worsening cytological grades. When comparing LSIL+ and ASC-US+,

A.  AICCS to diagnose cytology grade in ASC-US+

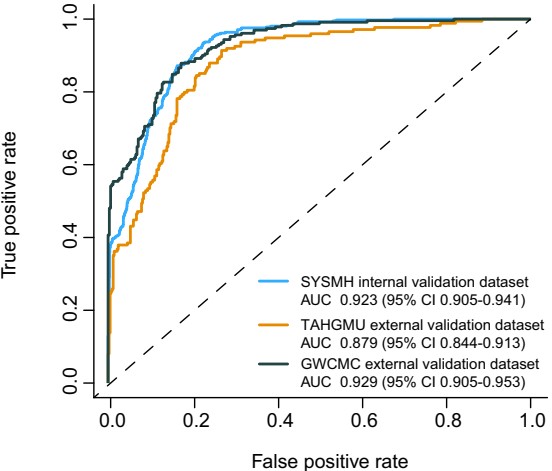

B.  AICCS to diagnose cytology grade in LSIL+

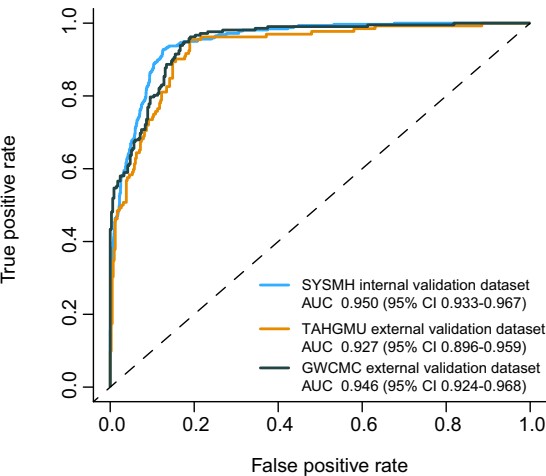

C.  AICCS to diagnose cytology grade in HSIL+

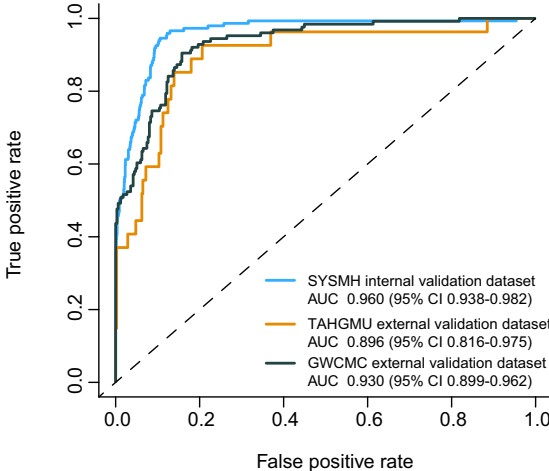

**Fig. 4 | ROC curves of AICCS system according to risk stratification obtained on different validation datasets.** ROC curves obtained by the AICCS system for participants with (**A**) ASC-US+, (**B**) LSIL+, and (**C**) HSIL+. The evaluation metric is ROC curves, with 95% confidence intervals in brackets. Source data are provided as a Source Data file.

LSIL+ maintained an advantage of 1.8% to 5.5% in both internal and external validation datasets. Additionally, comparing

HSIL+ (AUC: 0.960, 95% CI 0.938–0.982) to ASC-US+ (AUC: 0.923, 95% CI 0.905–0.941) in the internal validation dataset, we observed an advantage of up to 4%.

We also evaluated the performance of sensitivity, specificity, and accuracy in all subgroups (Fig. 5). In terms of sensitivity, HSIL+ exhibited advantages over other subgroups across all datasets. Additionally, the AICCS system showed comparable performance in specificity and accuracy among ASC-US+, LSIL+, and HSIL+, maintaining values above 0.810 for accuracy, and specificity across all internal and external validation datasets, regardless of different cytology grades. Furthermore, the AICCS system exhibited high negative predictive values (NPV) for the validation datasets, being 0.973 on the SYSMH internal validation dataset, 0.913 on the GWCMC external validation dataset, and 0.958 on the TAHGMU external validation dataset (Supplementary Table 7).

**Impact of diagnostic performance in cytopathologists with the assistance of AICCS in prospective validation dataset**

To assess the effectiveness of AICCS as a tool aiding cytopathologists in daily clinical practice, we evaluated the diagnostic performance using various approaches, including AICCS alone, cytopathologists, and AICCS-assisted cytopathologists, based on the prospective validation dataset from SYSMH. Table 1 demonstrates the high performance of sensitivity, specificity, accuracy, and AUC values in the AICCS alone, cytopathologists, and AICCS-assisted cytopathologists groups.

Compared to cytopathologists working without assistance, those aided by AICCS showed significant improvements in AUC (from 0.948 to 0.993; $P = 0.0006$), sensitivity (from 0.909 to 0.991; $P = 0.024$), specificity (from 0.987 to 0.996; $P = 0.002$), and accuracy (from 0.984 to 0.995; $P < 0.001$). Moreover, when compared to AICCS alone, AICCS-assisted cytopathologists exhibited significantly a higher AUC, specificity, and accuracy than the AICCS alone (all at $P < 0.001$) and maintained a comparable sensitivity ($P = 0.091$) across all cervical cytology grades.

Subgroup analyses, compared to AICCS alone (with AUC values of 0.947, 95% CI 0.936–0.958 for ASC-US+, 0.968, 95% CI 0.956–0.981 for LSIL+, and 0.965, 95% CI 0.949–0.982 for HSIL+), and cytopathologists with AUC values of 0.946, 95% CI(0.918–0.974 for ASC-US+, 0.975, 95% CI 0.949–1.000 for LSIL+, and 0.994, 95% CI 0.992–0.996 for HSIL+), AICCS-assisted cytopathologists (with AUC values of 0.993, 95% CI 0.984–1.000 for ASC-US+, 0.998, 95% CI 0.996–0.999 for LSIL+, and 0.998, 95% CI 0.996–0.999 for HSIL+) demonstrated significant improvements in AUC values, as indicated in Table 1 and Fig. 6. Additionally, AICCS-assisted cytopathologists demonstrated superiority in terms of specificity and accuracy (all $P < 0.05$) among patients with ASC-US+, LSIL+, or HSIL+. Moreover, the NPV was 0.997, 0.996, and 1.000 for AICCS alone, cytopathologists, and AICCS-assisted cytopathologists, respectively, as displayed in Supplementary Table 8.

To further evaluate the reliability of the AICCS, the performance comparison between AICCS alone and a group of cytopathologists was conducted using histopathological diagnosis as the gold standard. No significant difference was observed between AICCS alone and cytopathologists in detecting abnormal cytology grades. In subgroup analyses, both AICCS alone and cytopathologists showed high levels of sensitivity, particularly in the HSIL+ subgroup, where the sensitivity reached up to 1.000. This suggests that the AICCS system possesses robust capability in accurately identifying patients with epithelial lesion cells (LSIL, ASC-H, HSIL, SCC, and AGC), as presented in Supplementary Table 9.

A.  AICCS to diagnose all cervical cytology grades

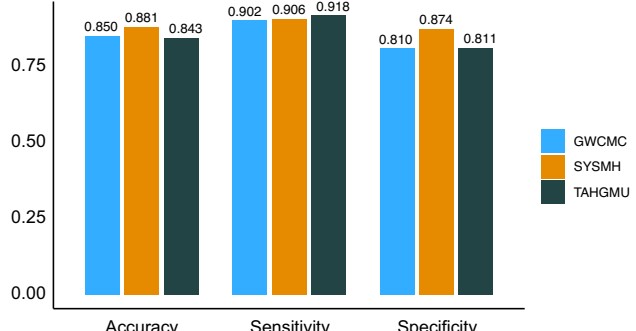

B.  AICCS to diagnose cytology grade in ASC-US+

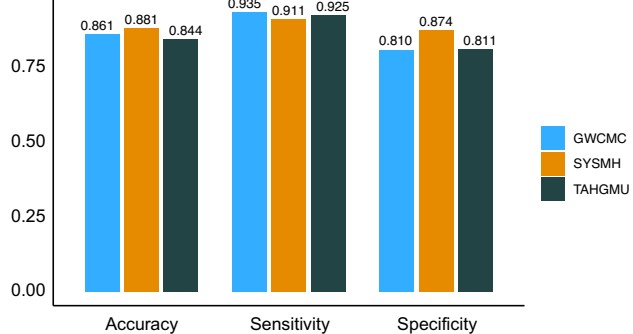

C.  AICCS to diagnose cytology grade in LSIL+

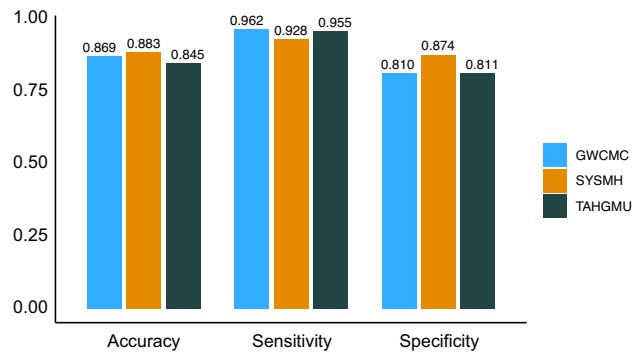

D.  AICCS to diagnose cytology grade in HSIL+

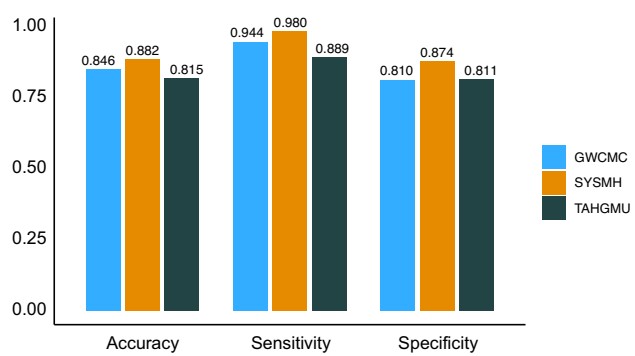

**Fig. 5 | Overall performance on retrospective validation datasets.** Overall performance of the AICCS system on all cytology grades among different validation datasets (**A**). Overall performance of the AICCS system on ASC-US+ among different validation datasets (**B**). Overall performance of the AICCS system on LSIL+ among different validation datasets (**C**). Overall performance of the AICCS system on HSIL+ among different validation datasets (**D**). Source data are provided as a Source Data file.

## Comparing the performance of different diagnostic measures in the randomized observational trial

Despite the AICCS system being developed based on a retrospective dataset, it showed high efficacy in identifying cervical cytology grades. To provide a more realistic assessment of its effectiveness, we conducted a randomized observational trial at SYSMH to compare the efficacy of AICCS alone, cytopathologists, and AICCS-assisted cytopathologists. All three groups exhibited high AUC values, consistent with findings from prospective validation. For participants with ASC-US+, the AUC values were 0.944, 0.959, and 0.995, respectively (Fig. 7A). Furthermore, for participants with LSIL+ and HSIL+, AUCs exceeding 0.980 were achieved (Fig. 7B, C). In comparison to the AICCS alone, AICCS-assisted cytopathologists showed superiority in terms of specificity and accuracy among patients with ASC-US+, LSIL+, or HSIL+ (all $P < 0.001$), while maintaining comparable sensitivity for abnormal cytology grades ($P > 0.05$) (Table 2, Supplementary Fig. 3). The NPV was 0.994, 0.989, and 1.000 for AICCS alone, cytopathologists, and AICCS-assisted cytopathologists, respectively (Supplementary Table 8).

## Clinical application of the AICCS system

In addition to enhancing overall diagnostic performance, the AICCS system also accelerates the diagnostic process. The time required for AICCS analysis of one WSI was under 120 s, whereas manual reading took approximately 180 s.

To facilitate AICCS-assisted diagnosis and consultations for complex cases in clinical settings, a cloud-based, multi-institutional AI platform was developed. Clinicians can access this platform through a publicly accessible website (https://ai-eng.cellsvision.com:3443/), enabling them to upload cervical cytology smears for analysis

(Supplementary Fig. 4). Experimental implementation is currently underway at multiple institutions. We are confident that the AICCS system holds the potential to enhance medical services and advance clinical deployment (Supplementary Fig. 5).

## Discussion

This study aimed to develop and validate an AICCS system for diagnosing cervical cytology classifications by analyzing cervical cell WSIs. The AICCS system underwent training and testing on diverse datasets made up of 16,056 participants. It employed two AI models: one for cell detection and another for WSI classification. The research utilized multicenter, retrospective, and prospective population datasets, along with a randomized observational trial for system validation. The validation across these datasets revealed the AICCS system's outstanding performance in cervical cytology screening and differential classifications.

In our prospective evaluation, the proposed system achieved an AUC of 0.947, sensitivity of 0.946, specificity of 0.890, and accuracy of 0.892. Particularly noteworthy, in the randomized observational trial, the AICCS-assisted cytopathologists surpassed cytopathologists alone, exhibiting significantly higher AUC, specificity, and accuracy, along with a notable 13.3% improvement in sensitivity. The AICCS system showed exhibited promise as an adjunct tool for precise and efficient cervical cancer screening. Furthermore, promising outcomes were observed for the AICCS system in accurately identifying cervical cytology grading. The system demonstrated high sensitivity and specificity, indicating its capability to detect abnormal cells and effectively differentiate them from normal cells. Therefore, this system can potentially lead to earlier detection and intervention for cervical cancer, thereby enhancing patient outcomes and alleviating the burden on health-care systems.

**Table 1 | Performance of the AICCS alone, cytopathologists, and AICCS-assisted cytopathologists in the SYSMH prospective validation datasets**

| | AICCS alone | Cytopathologists | AICCS-assisted cytopathologists | AICCS alone vs Cytopathologists P value | AICCS alone vs AICCS-assisted P value | AICCS-assisted vs Cytopathologists P value |
|---|---|---|---|---|---|---|
| **All cervical cytology grades** | | | | | | |
| Sensitivity (95% CI) | 0.946 (0.885–0.980) | 0.909 (0.839–0.956) | 0.991 (0.950–1.000) | 0.304 | 0.091 | 0.024 |
| Specificity (95% CI) | 0.890(0.878–0.902) | 0.987(0.982–0.991) | 0.996 (0.992–0.998) | <0.001 | <0.001 | 0.002 |
| Accuracy (95% CI) | 0.892 (0.880–0.904) | 0.984 (0.979–0.989) | 0.995 (0.992–0.998) | <0.001 | <0.001 | <0.001 |
| AUC (95% CI) | 0.947 (0.936–0.958) | 0.948 (0.921–0.975) | 0.993 (0.984–1.000) | 0.952 | <0.001 | 0.0006 |
| **Stratification analysis** | | | | | | |
| **NILM** | | | | | | |
| Sensitivity (95% CI) | 0.890 (0.878–0.902) | 0.989 (0.984–0.992) | 0.996 (0.992–0.998) | <0.001 | <0.001 | 0.024 |
| Specificity (95% CI) | 0.953 (0.893–0.985) | 0.906 (0.833–0.954) | 0.991 (0.949–1.000) | 0.024 | 0.136 | 0.024 |
| Accuracy (95% CI) | 0.893 (0.881–0.904) | 0.986 (0.980–0.990) | 0.995 (0.992–0.998) | <0.001 | <0.001 | <0.001 |
| AUC (95% CI) | 0.947 (0.936–0.958) | 0.946 (0.918–0.974) | 0.993 (0.984–1.000) | 0.963 | <0.001 | 0.0006 |
| **ASC-US+** | | | | | | |
| Sensitivity (95% CI) | 0.953 (0.893–0.985) | 0.906 (0.833–0.954) | 0.991 (0.949–1.000) | 0.189 | 0.136 | 0.024 |
| Specificity (95% CI) | 0.890 (0.878–0.902) | 0.989 (0.984–0.992) | 0.996 (0.992–0.998) | <0.001 | <0.001 | 0.007 |
| Accuracy (95% CI) | 0.893 (0.881–0.904) | 0.986 (0.980–0.990) | 0.995 (0.992–0.998) | <0.001 | <0.001 | <0.001 |
| AUC (95% CI) | 0.947 (0.936–0.958) | 0.946 (0.918–0.974) | 0.993 (0.984–1.000) | 0.963 | <0.001 | 0.0006 |
| **LSIL+** | | | | | | |
| Sensitivity (95% CI) | 0.981 (0.899–1.000) | 0.962 (0.870–0.995) | 1.000 (0.933–1.000) | 0.566 | 0.497 | 0.291 |
| Specificity (95% CI) | 0.890 (0.878–0.902) | 0.989 (0.984–0.992) | 0.996 (0.992–0.998) | <0.001 | <0.001 | 0.007 |
| Accuracy (95% CI) | 0.892 (0.880–0.903) | 0.988 (0.983–0.992) | 0.996 (0.992–0.998) | <0.001 | <0.001 | 0.004 |
| AUC (95% CI) | 0.968 (0.956–0.981) | 0.975 (0.949–1.000) | 0.998 (0.996–0.999) | 0.682 | <0.001 | 0.083 |
| **HSIL+** | | | | | | |
| Sensitivity (95% CI) | 1.000 (0.877–1.000) | 1.000 (0.877–1.000) | 1.000 (0.877–1.000) | NA | NA | NA |
| Specificity (95% CI) | 0.890 (0.878–0.902) | 0.989 (0.984–0.992) | 0.996 (0.992–0.998) | <0.001 | <0.001 | 0.007 |
| Accuracy (95% CI) | 0.891 (0.879–0.903) | 0.989 (0.984–0.993) | 0.996 (0.992–0.998) | <0.001 | <0.001 | 0.007 |
| AUC (95% CI) | 0.965 (0.949–0.982) | 0.994 (0.992–0.996) | 0.998 (0.996–0.999) | 0.001 | <0.001 | <0.001 |

All cervical cytology grades include NILM, ASC-US, LSIL, ASC-H, HSIL, SCC, and AGC. ASC-US+ includes ASC-US, LSIL, ASC-H, HSIL and SCC. LSIL+ includes LSIL, ASC-H, HSIL, and SCC. HSIL+ includes ASC-H, HSIL, and SCC. The χ2 test (two-sided) was used for two-group categorical variables. Statistical significance was considered when the two-tailed P value was less than 0.05.

A.  Performance of AICCS and cytopathologist in ASC-US+

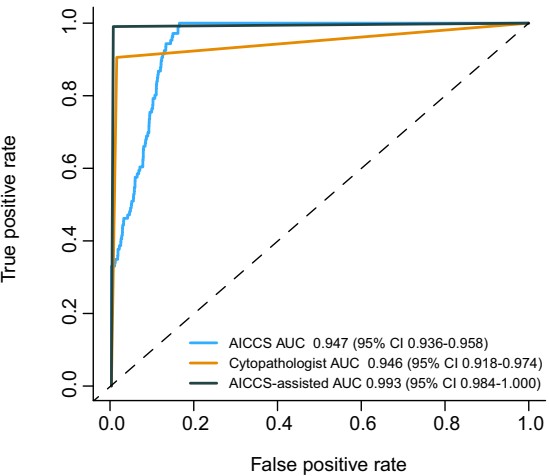

A.  Performance of AICCS and cytopathologist in ASC-US+

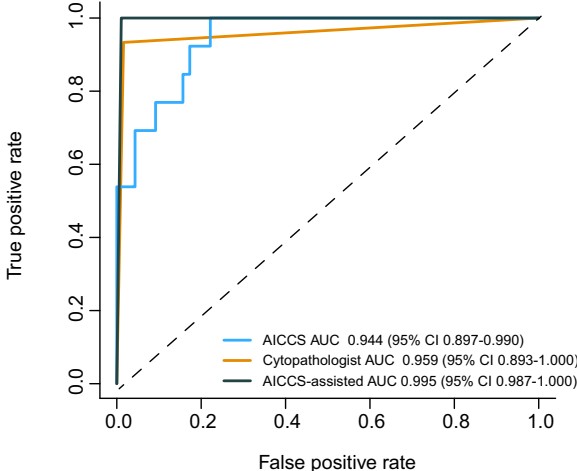

B.  Performance of AICCS and cytopathologist in LSIL+

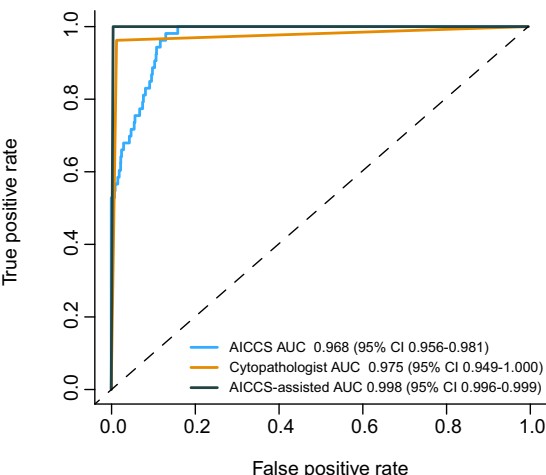

B.  Performance of AICCS and cytopathologist in LSIL+

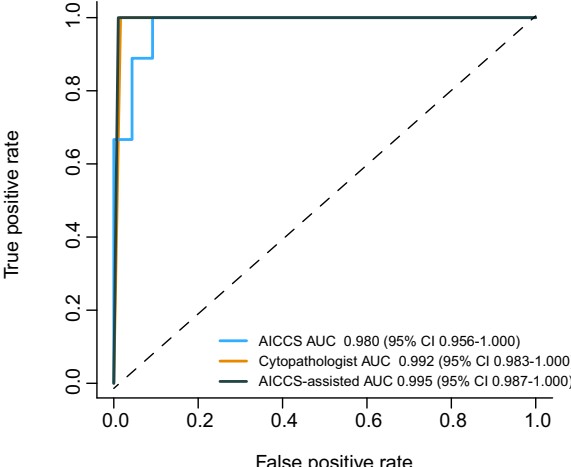

C.  Performance of AICCS and cytopathologist in HSIL+

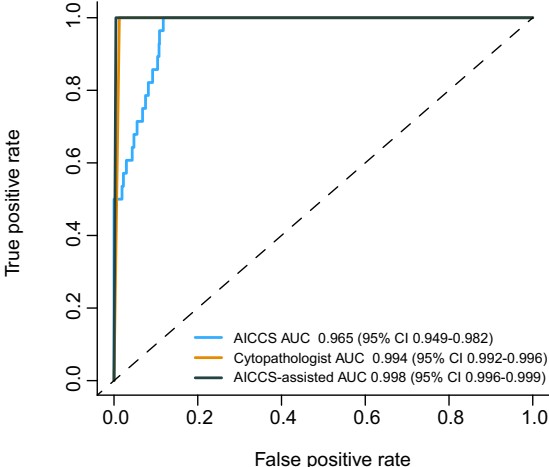

C.  Performance of AICCS and cytopathologist in HSIL+

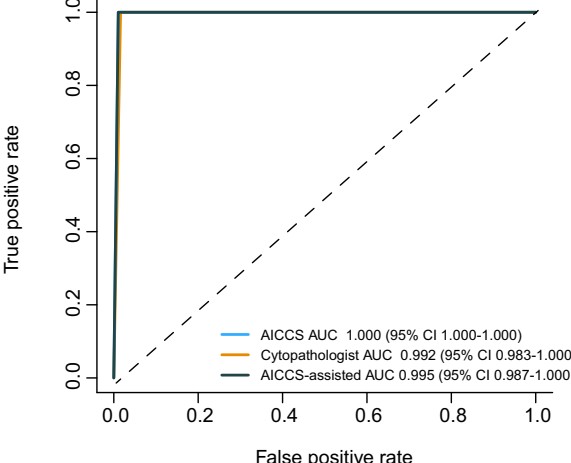

**Fig. 6 | Comparisons of diagnostic performance of the AICCS alone, cytopathologists, and AICCS-assisted cytopathologists on the prospective validation dataset.** Participants with (**A**) ASC-US+, (**B**) LSIL+, and (**C**) HSIL+. The evaluation metric is ROC curves, with 95% confidence intervals in brackets. Source data are provided as a Source Data file.

**Fig. 7 | Comparisons of diagnostic performance of the AICCS alone, cytopathologists, and AICCS-assisted cytopathologists in the randomized observational trial.** Participants with (**A**) ASC-US+, (**B**) LSIL+, and (**C**) HSIL+. The evaluation metric is ROC curves, with 95% confidence intervals in brackets. Source data are provided as a Source Data file.

**Table 2 | Performance of the AICCS alone, cytopathologists, and AICCS-assited cytopathologists in cervical cytopathological diagnosis in the randomized observational trial**

| | AICCS alone (N = 201) | cytopathologists (N = 203) | AICCS-assisted cytopathologists (N = 204) | AICCS alone vs cytopathologists P value | AICCS alone vs AICCS-assisted P value |
|---|---|---|---|---|---|
| **All cervical cytology grades** | | | | | |
| Sensitivity (95% CI) | 0.938 (0.698–0.998) | 0.867 (0.595–0.983) | 1.000 (0.782–1.000) | 0.552 | 0.487 |
| Specificity (95% CI) | 0.854 (0.795–0.902) | 0.984 (0.954–0.997) | 0.989 (0.962–0.999) | <0.001 | <0.001 |
| Accuracy (95% CI) | 0.861 (0.805–0.905) | 0.975 (0.943–0.992) | 0.990 (0.965–0.999) | <0.001 | <0.001 |
| AUC (95% CI) | 0.932 (0.889–0.976) | 0.929 (0.845–1.000) | 0.995 (0.987–1.000) | NA | NA |
| **Stratification analysis** | | | | | |
| **NILM** | | | | | |
| Sensitivity (95% CI) | 0.854 (0.795–0.902) | 0.984 (0.954–0.997) | 0.989 (0.962–0.999) | <0.001 | 0.378 |
| Specificity (95% CI) | 0.923 (0.640–0.998) | 0.929 (0.661–0.998) | 1.000 (0.715–1.000) | 0.916 | 0.103 |
| Accuracy (95% CI) | 0.857 (0.802–0.904) | 0.980 (0.950–0.995) | 0.990 (0.964–0.999) | <0.001 | <0.001 |
| AUC (95% CI) | 0.944 (0.897–0.990) | 0.959 (0.893–1.000) | 0.995 (0.987–1.000) | NA | NA |
| **ASC-US+** | | | | | |
| Sensitivity (95% CI) | 0.923 (0.640–0.998) | 0.929 (0.661–0.998) | 1.000 (0.715–1.000) | 1 | 0.013 |
| Specificity (95% CI) | 0.854 (0.795–0.902) | 0.984 (0.954–0.997) | 0.989 (0.962–0.999) | <0.001 | <0.001 |
| Accuracy (95% CI) | 0.859 (0.802–0.904) | 0.980 (0.950–0.995) | 0.990 (0.964–0.999) | <0.001 | <0.001 |
| AUC (95% CI) | 0.944 (0.897–0.990) | 0.959 (0.893–1.000) | 0.995 (0.987–1.000) | NA | NA |
| **LSIL+** | | | | | |
| Sensitivity (95% CI) | 1.000 (0.664–1.000) | 1.000 (0.541–1.000) | 1.000 (0.541–1.000) | NA | NA |
| Specificity (95% CI) | 0.854 (0.795–0.902) | 0.984 (0.954–0.997) | 0.989 (0.962–0.999) | <0.001 | <0.001 |
| Accuracy (95% CI) | 0.861 (0.804–0.906) | 0.985 (0.955–0.997) | 0.990 (0.963–0.999) | <0.001 | <0.001 |
| AUC (95% CI) | 0.980 (0.956–1.000) | 0.992 (0.983–1.000) | 0.995 (0.987–1.000) | NA | NA |
| **HSIL+** | | | | | |
| Sensitivity (95% CI) | 1.000 (0.025–1.000) | 1.000 (0.292–1.000) | 1.000 (0.478–1.000) | NA | NA |
| Specificity (95% CI) | 0.854 (0.795–0.902) | 0.984 (0.954–0.997) | 0.989 (0.962–0.999) | <0.001 | <0.001 |
| Accuracy (95% CI) | 0.855 (0.796–0.902) | 0.984 (0.955–0.997) | 0.990 (0.963–0.999) | <0.001 | <0.001 |
| AUC (95% CI) | 1.000 (1.000–1.000) | 0.992 (0.983–1.000) | 0.995 (0.987–1.000) | NA | NA |

All cervical cytology grades include NILM, ASC-US, LSIL, ASC-H, HSIL, SCC, and AGC. ASC-US+ includes ASC-US, LSIL, ASC-H, HSIL, SCC. LSIL+ includes LSIL, ASC-H, HSIL, and SCC. HSIL+ includes ASC-H, HSIL, and SCC. The χ2 test (two-sided) was used for two-group categorical variables. Statistical significance was considered when the two-tailed P value was less than 0.05.

In general, the clinical applications of AI are limited owing to a lack of validation in prospective datasets[22]. However, this study stands out as the AICCS system was trained using over 10,000 smears and validated using two independent external hospital datasets, in addition to a prospective validation dataset. Furthermore, a randomized observational trial was conducted to demonstrate the high sensitivity and accuracy of both the AICCS alone and the AICCS-assisted cytopathologists.

Despite the approval of a few Class-II and Class-III Cervical AI-assisted analysis software by the National Medical Products Administration (NMPA) of China in 2023, China currently still lacks an effective cervical cancer screening system. Therefore, it is crucial to establish an AI cervical cancer screening system tailored to the country's specific conditions. In China, public medical services are typically organized at the provincial, municipal, and county levels. However, many samples collected from rural areas or lower-tier cities in developing regions are processed at county hospitals, where skilled cytopathologists are scarce. The system developed in this study presents a potentially affordable solution. Additionally, there are multiple options available for providing AI-assisted diagnostic services, such as on-premises, software-as-a-service (SaaS), and pay-per-slide models. The combination of 5G networks and intelligent medicine can help address the resource imbalance in health-care and enhance the quality of medical services[23–26].

However, several limitations need to be considered. Firstly, owing to limited operator experience, not every sample may accurately represent the true state of cervical lesions, potentially leading to false-negative events. Secondly, variations in skill levels among liquid-based cytology preparers across medical centers could result in inconsistent quality in the prepared smears, thereby affecting the accuracy of the diagnostic results. Thirdly, although SCC hold significant diagnostic value, they are not explicitly delineated as a separate WSI category, owing to the limited number of samples. Additionally, ethical considerations must be addressed when implementing an AI-based screening system. The potential impact on patient privacy, data security, and the role of health-care professionals in decision-making processes should be carefully evaluated and monitored.

In conclusion, our study highlights showcase the potential of AI-assisted cervical cytology screening, with our proposed AICCS system demonstrating remarkable diagnostic accuracy and the capability to assist health-care professionals. This technology holds the potential to significantly enhance cervical cancer detection and clinical practice, thereby paving the way for improved health-care services and deployment in medical institutions.

## Methods

### Study design and participants

Between January 2016 and December 2020, a total of 16,056 eligible participants were enrolled in this multicenter study. Distinct datasets were created, including retrospective and prospective population-based datasets, as well as a randomized observational trial, to train and validate the AICCS system for the auxiliary diagnosis of cervical cytology grade and cervical cancer. The AICCS system consists of two main functional AI models: a patch-level cell detection model and a WSI-level classification model. Our study adhered to the reporting and analysis guidelines of STARD, CONSORT-AI Extension, and the MI-CLAIM checklist[27–29]. This study obtained approval from the institutional review boards of each participating hospital and adhered to the principles outlined in the Declaration of Helsinki.

The AICCS system underwent training and validation for cervical cancer screening in three phases. In the proof-of-concept (POC) phase, the training phase involved the retrospective acquisition of WSIs. In the validation phase, the AICCS underwent internal and external validation using multicenter, retrospective, and prospective population-based datasets. The third phase involved further validation through a randomized observational trial. Supplementary Figs. 1 and 2 present the design flowchart of the study.

### WSI acquisition and preprocessing

For the acquisition and preprocessing of WSIs, cervical liquid-based preparation samples collected and maintained using the sedimentation liquid-based preparation method were initially digitalized. WSIs were generated using two prominent digital pathology scanners manufactured in China. One scanner was the PRECICE 600 (UNIC TECHNOLOGIES, INC.), equipped with a 40× objective lens, providing a specimen-level pixel size of 0.2529 μm × 0.2529 μm. The second scanner was the KF-PRO-400-HI (Ningbo Jiangfeng Bio-Information Technology Co., Ltd.), also featuring a 40× objective lens and offering a specimen-level pixel size of 0.2484 μm × 0.2484 μm. Depending on the size of the smear sample, each WSI (scanned at 40× objective power) contained billions of pixels and had a data size ranging from several hundred megabytes to a few gigabytes. All WSIs were saved in a proprietary confidential format.

Six cytopathologists from the Cellular and Molecular Diagnostics Center, SYSMH, each with over 5 years of experience, participated in cell annotation and WSI classification according to The Bethesda System (TBS) 2014 guidelines. Negative smears did not require annotation or labeling. As per TBS 2014 criteria, satisfactory samples were defined as those containing a minimum of 5000 visible and uncovered squamous epithelial cells with the presence of abnormal cells (atypical squamous cells or atypical glandular cells and above). Samples with fewer than 5000 visible cells, uncovered squamous epithelial cells, or those affected by blood, inflammatory cells, epithelial cell overlapping, poor fixation, excessive drying, or unknown component contamination affecting over 75% of squamous epithelial cells were excluded.

Quality control was ensured through participant eligibility assessment and adherence to strict specimen criteria. Participant inclusion criteria comprised being 18 years or older, not pregnant, and free from mental illness or cognitive impairment. All participants provided consent for cervical liquid-based cytology for definitive diagnosis. In addition to manual quality control measures, we implemented an AI-assisted approach to identify and address potential scanning quality issues during the digitization process. For this purpose, we developed an image classification model utilizing thumbnail images of WSIs to detect instances of scanning quality hindrances such as blurriness or incomplete scanning areas (Fig. S6). The thumbnails' short edges were standardized to 1000 pixels, to represent scaled-down versions of the original WSIs. Images flagged by this model as potentially problematic underwent manual review. Upon confirmation of scanning quality issues, they were then excluded from our dataset.

A quality assessment model was constructed with three main components: an EfficientNet backbone, a quality summary branch, and a quality detail branch. The EfficientNet backbone was employed for feature extraction. The quality summary branch provided an overall assessment of slide quality, performing binary classification to categorize slides as either acceptable or problematic. In contrast, the quality detail branch offered a more nuanced evaluation by estimating the severity of specific quality issues. It employed a multi-label classification technique to identify and categorize different types of quality problems within the slides.

### ROI annotation and WSI labeling

All cervical liquid-based preparation samples underwent review by cytopathologists from the Cellular and Molecular Diagnostics Center, SYSMH. with unsatisfactory samples being excluded. Following digitization, each WSI was randomly assigned to two of the six cytopathologists for independent annotation and labeling. In cases where consensus was not reached, an expert cytopathologist conducted further review. During the patch-level annotation phase, 2848 WSIs served as training data for the patch-level deep neural network

detection model. Each pathologist was randomly allocated 950 WSIs and tasked with marking and labeling positive cells within each WSI's bounding box, while adhering to TBS 2014. For the WSI-based classification model construction phase, 9,316 WSIs from the SYSMH training dataset were diagnosed and labeled by the cytopathologists (Supplementary Fig. 1). Approximately 3106 WSIs were equally randomized to each cytopathologist for labeling.

Based on the TBS 2014 guidelines, classification included both NILM and epithelial lesions. Epithelial lesions included squamous epithelial lesions and glandular epithelial lesions. Moreover, squamous epithelial lesions were further categorized into ASC-US, LSIL, ASC-H, HSIL, and SCC. Glandular epithelial lesions included atypical glandular cells, not otherwise specified (AGC-NOS), atypical glandular cells, favor neoplastic (AGC-FN), endocervical adenocarcinoma in situ (AIS), and adenocarcinoma (ADC). A list of all abbreviations and classifications are provided in Supplementary Table 10.

Patch-level annotation of WSIs encompassed comprised two distinct phases: an initial manual annotation phase and an AI-suggested annotation phase. In the initial manual annotation phase, cytopathologists participated in labeling a subset of patches, wherein all glandular epithelial lesions were grouped into the category of atypical glandular cells (AGC). This categorization was due to the low detection rate, limited specimen quantities, overlapping morphological features, and similar clinical management approaches associated with AGC. Thus, cytopathologists annotated six categories of positive cells considered highly representative or typical: ASC-US, LSIL, ASC-H, HSIL, SCC, and AGC. It is important to note that negative smears did not require annotation. Subsequent to the initial phase of manual annotation, a detection model was trained and deployed to perform a sliding-window inference on WSIs, thereby generating AI-suggested regions of interest (ROIs), encompassing cells identified as intraepithelial lesions (Supplementary Fig. 7). These AI-suggested ROIs underwent review and annotation by cytopathologists before integration into our patch-level training dataset. This iterative process, involving the confirmation and potential adjustment of AI-recommended ROIs by cytopathologists, ensured a progressive refinement of AI performance based on expert cytopathological input. Ultimately, this procedure methodology educated the model to discern abnormal cells within WSIs, each potentially containing tens of thousands of cells. The output generated by the cell detection model served as input for the WSI-level classification models, highlightng its crucial role in the overarching analytical framework. This approach facilitated the construction of our training dataset with high-quality annotations while expediting the annotation process.

For WSI-level classification, patches annotated as ASC-H, HSIL, and SCC were grouped into the WSI-level category of HSIL+ owing to their morphological similarities and similar clinical management. Consequently, WSI-level classifications comprised five categories: NILM, ASC-US, LSIL, HSIL+, and AGC. The detailed procedures for classification at both the patch-level and WSI-level are presented in Supplementary Fig. 8.

Data augmentation, particularly color augmentation, was systematically applied rather than randomly or blindly in order to play a crucial role in improving the accuracy of deep learning object detection frameworks and addressing overfitting concerns. Augmentation strategies included various techniques such as random patch cuts around annotated cells with different overlapping ratios, random rotations, and alterations in staining colors. The specific steps encompassed within this augmentation procedure were as follows: Before training, the distribution of the H and E components for all images in the training set was quantified, and their mean values ($\mu$), standard deviations ($\sigma$), as well as upper and lower bounds were fitted. During training, the H and E components of the current image were first obtained. Then, either the H component, the E component, both, or neither were randomly selected, and random sampled within the range of $[-2\sigma, 2\sigma]$ for the corresponding component were uniformly sampled. These numbers were added to the original components of the image, ensuring that the values would remain within the specified upper and lower limits after the operation. In summary, each RGB image was transformed into the stain density absorbance (SDA) space. Subsequently, the Macenko method was utilized for perform color deconvolution, resulting in a $3 \times 3$ stain component matrix. Finally, the perturbed stain component matrix was employed to reconstruct the SDA image back into the RGB space. This approach ensured that the color augmentation process was both controlled and consistent with common practices in pathological image processing, thereby mitigating potential performance variations arising from staining differences.

## Development and architecture of the AICCS system for cervical cytology diagnosis

The development and architecture of the AICCS system involved training and validating the system using retrospectively obtained images from 11,468 eligible individuals at SYSMH. These images were randomly divided into a training cohort ($n = 9316$) and an internal validation cohort ($n = 2152$) at a 4:1 ratio. The AICCS system consisted of two major functional models: a patch-level cell detection model and a WSI-level classification model.

During the operational mode, the trained cell detection model processed a WSI by dividing it into smaller patches using a sliding window approach. Subsequently, the model annotated abnormal cells based on the criteria defined by TBS 2014. The output of the cell detection model serves as input for the WSI-level classification model.

The WSI-level classification model utilizes the results from the patch-level cell detection model and employed a set of well-designed features. These features were then fed into the trained WSI-level classification model, which assigns one of five possible cytology grades according to TBS 2014: NILM, ASC-US, LSIL, HSIL+, and AGC. The workflow chart of the AICCS is depicted in Fig. 1.

## Validation and comparative analysis of the AICCS system: retrospective and prospective evaluation

The performance of the AICCS system in predicting cervical cytology images was initially retrospectively validated on the SYSMH internal validation dataset ($n = 2152$). Subsequently, external validation was conducted using the GWCMC dataset ($n = 600$) and TAHGMU dataset ($n = 600$).

To further evaluate the generalizability and robustness of the AICCS system in clinical practice, cervical cytology images from 2780 eligible participants were prospectively collected at SYSMH. The same cervical cytology images were individually reviewed by cytopathologists, comparing their results with those obtained solely from the AICCS system and those obtained by cytopathologists assisted by the AICCS system.

In the cytopathologist group, digital cervical images were individually screened by cytopathologists, and the TBS class for each image was determined without any additional assistance. In the AICCS group, screening results were automatically generated by the AICCS alone. In the AICCS-assisted cytopathologists group, the AICCS was first employed to screen the WSI and generate potential abnormal cell boxes. Then, a second screening was conducted by a cytopathologist before the final decision was made.

## Evaluation and randomized observational trial design

To compare the performance of cytopathologists, the AICCS alone, and AICCS-assisted cytopathologists, a randomized observational trial was conducted at SYSMH from August 13, 2020, to December 14, 2020. A total of 608 participants who met our inclusion criteria were randomly assigned in a 1:1:1 ratio to receive a diagnosis from cytopathologists, the AICCS alone, or AICCS-assisted cytopathologists. There were no withdrawals from the study after randomization. To prevent

selection bias, randomization was performed using a random number generator without any stratification factors. After receiving an initial diagnosis, all participants, identified only by a masked identification number, received a gold-standard diagnosis from an expert cytopathologist.

## AICCS algorithms

Deep learning-based object detection models fall into two primary categories: two-stage detectors and one-stage detectors. The two-stage object detection approach involves generating region proposals, followed by classifying and refining these proposals. The R-CNN family of object detectors, including R-CNN, Fast-R-CNN, and Faster-R-CNN[30], gained significant popularity and were state-of-the-art object detection models for an extended period. In contrast, the one-stage object detection approach can directly predict objects without the intermediary step of region proposal generation. One-stage object detection methods aim to streamline the object detection pipeline by predicting object class labels and bounding box coordinates in a single pass. This often results in faster processing speeds compared with two-stage methods. Owing to their efficiency, they have become favored options for real-time object detection. Prominent examples of one-stage object detection models include YOLO, SSD, and RetinaNet[31].

In the proposed AICCS system, based on our selection studies for the two major detection approaches (Supplementary Tables 1, 2), we adopted RetinaNet, a one-stage object detection approach, as our anomaly cell detection model at the patch-level. RetinaNet leverages convolutional neural networks (CNNs) and employs ResNet+FPN (ResNet plus a feature pyramid network) as its backbone for feature extraction. Additionally, it incorporates two task-specific subnetworks for classification and bounding box regression. Notably, RetinaNet introduces a focal loss function to address the class imbalance between the foreground and background, which is a common issue in medical data. It thus achieves good performance. ResNet, with its identity shortcut connection that bypasses one or more layers, mitigates the vanishing gradient problem in deep neural networks. Furthermore, the FPN enhances a standard convolutional network by incorporating a top-down pathway and lateral connections, thereby efficiently constructing a multi-scale feature pyramid from a single-resolution input image.

Considering the TBS 2014 guidelines and real-world data distribution, we designed our patch-level detector to distinguish between six classes of abnormal cells: ASC-US, LSIL, ASC-H, HSIL, SCC, and AGC. Owing to the distinct morphological patterns between squamous cells and glandular cells, our patch-level detector integrates an additional binary classifier subnetwork to distinguish between squamous epithelial cells and glandular cells. This subnetwork complements the conventional subnetworks responsible for multiclass recognition and position localization, thereby allowing for the incorporation of loss from all subnetworks during the training process.

The output of the detection model served two main purposes as follows: (1) The top 20 detections of each class were displayed in the graphic user interface of the AICCS system, enabling cytopathologists to review and select them as evidence when issuing a final report. (2) All detections are aggregated at the WSI level and utilized as input for the WSI-level classifier.

Furthermore, in the WSI process, the foundational step in developing a WSI classification model is to generate features based on statistical data from the WSI detection results. Initially, the development of a WSI classification model begins with generating of features that encapsulate the statistical data derived from the WSI detection outcomes. Specifically, these statistical metrics encompass the distribution of confidence levels for each classified object, encompassing the maximum, mean, and standard deviation of the confidence scores for objects within each category, as well as the proportion of each confidence interval pertaining to a given class. Subsequently, these statistical metrics are subsequently converted into features, which are

then utilized to train the WSI classification model via the implementation of a random forest algorithm. The top 20 features selected in the random forest model areillustrated in Supplementary Fig. 9.

## Statistical analysis

The performance of cytopathologists, the AICCS alone, and AICCS-assisted cytopathologists in identifying cervical cytology grades was evaluated by determining the sensitivity, specificity, accuracy, and area under the curve (AUC) for each group. Receiver operating characteristic (ROC) curves were then plotted to visually demonstrate the diagnostic ability of cytopathologists, the AICCS alone, and AICCS-assisted cytopathologists in classifying cervical cytology grades.

To compare continuous variables, an independent $t$ test was conducted, and a $\chi^2$ test was employed for two-group categorical variables. The $P$ value and 95% confidence interval (CI) were utilized to compare the performance of cytopathologists, the AICCS alone, and AICCS-assisted cytopathologists in determining cervical cytology grades.

Statistical significance was considered when the two-tailed $P$ value was less than 0.05 for all statistical tests. Model training and validation were conducted using Python (version 3.6.8). Statistical analyses were performed using Python (version 3.6.8) and Medcala (version 15).

## Reporting summary

Further information on research design is available in the Nature Portfolio Reporting Summary linked to this article.

## Data availability

The datasets are governed by data usage policies specified by the data controller (Sun Yat-sen Memorial Hospital, Sun Yat-sen University). The WSIs, codes and expected output involved in the main text are securely maintained by the Ethics Committee of Sun Yat-sen Memorial Hospital. Source data are provided with this paper. This minimum dataset and source data file, have been upload to zenodo, the DOI is https://doi.org/10.5281/zenodo.10828395.

## Code availability

In this study, the digital scanner (PRECICE 600 (UNIC TECHNOLOGIES, INC.), KF-PRO-400-HI (Ningbo Jiangfeng Bio-Information Technology Co., Ltd.)) specific data reading SDK packages under commercial license were used for WSI importing. The networks used in our AICCS system were developed in Python (version 3.6). Our patch level abnormal cell detection was based on RetinaNet (https://github.com/jkznst/RetinaNet-mxnet, an unofficial implementation of ICCV 2017 RetinaNet (Focal Loss)). The WSI level classification algorithm was based on Random Forest (scikit-learn 0.23.2). The related codes are stored in https://github.com/cellsvision/AICCS, and now is linked to zenodo https://doi.org/10.5281/zenodo.10847469.

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

## Acknowledgements

We would like to thank for Cells Vision (Guangzhou) Medical Technology Inc. for help and support on the development of the AICCS system. And We thank Dr. Jennifer Geara from Karolinska Institutet for supporting this study. This study was supported by grants 2023YFE0204000 from National Key R&D Program of China, grants 2023B1212060013 from the Science and Technology Planning Project of Guangdong Province, grants 82273204 from the National Natural Science Foundation of China, grant 2023A1515012412 GuangDong Basic and Applied Basic Research Foundation, grant 202206010078, and 202201020574 from the Guangzhou Science and Technology Project, grant 2018007 from the Sun Yat-Sen University Clinical Research 5010 Program, grant SYS-C-201801 from the Sun Yat-Sen Clinical Research Cultivating Program, grant A2020558 from the Guangdong Medical Science and Technology Program, grant 7670020025 from Tencent Charity Foundation, and the ownership of these grants is attributed to H.Y. This study was supported by grant 2023A1515011214 GuangDong Basic and Applied Basic Research Foundation, grants 2023A03J0722 and 2024A03J1194 Guangzhou Science and Technology Project, grants YXQH202209 and SYSQH-II-2024-07 the Scientific Research Launch Project of Sun Yat-Sen Memorial Hospital, and the ownership of these grants is attributed to Y.Y.

## Author contributions

All authors have full access to all of the data in the study and take responsibility for the integrity of the data and accuracy of the data analysis. These authors contributed equally: Jue Wang, Yunfang Yu, Yujie Tan, Huan Wan, and Nafen Zheng. These authors jointly supervised this work: Herui Yao, Nengtai Ouyang, and Jin Wang are co-corresponding authors. Concept and design: Herui Yao, Nengtai Ouyang, Yufang Yu, Jue Wang, Yujie Tan, Huan Wan, and Nafen Zheng. Acquisition: Jue Wang, Huan Wan, Nafen Zheng, Gui He, Sha Fu, Yang Song, Qinyue Chen, Linna Zuo, Liya Wei, Ruichao Chen, and Hui Xu. Review cervical liquid-based preparation samples: Jue Wang, Huan Wan, Nafen Zheng, Sha Fu, Gui He, and Yang Song. Analysis, or interpretation of data: Jin Wang, Zhen Lin, Kai Liu, and Qinyue Yao. Drafting of the manuscript: All authors. Critical revision of the manuscript for important intellectual content: All authors. Statistical analysis: Jin Wang, Zhen Lin, Kai Liu, and Qinyue Yao. Obtained funding: Herui Yao. Administrative, technical, or material support: Herui Yao, Nengtai Ouyang, Jue Wang, Yufang Yu, Yujie Tan, Huan Wan, and Nafen Zheng. Supervision: Herui Yao, Nengtai Ouyang, and Jin Wang.

## Competing interests

The authors declare no competing interests.

## Consent to publish

The current study did not report individual participant data in any form, so informed consent was not applicable. All authors reviewed the manuscript and consented for publication.

## Ethics approval and consent to participate

This study obtained approval from the institutional review boards of each participating hospital and adhered to the principles outlined in the Declaration of Helsinki. Informed patient consent was waived retrospectively by the three study institutes and prospectively for the

prospective validation dataset, as all samples were irreversibly anonymized by the institutional review boards. Participants in the randomized observational trial provided written informed consent prior to their involvement. The study was registered on ClinicalTrials.gov (Number NCT04551287).
