## [Peer Review File · Nature Communications]

Artificial Intelligence Enables Precision Diagnosis of Cervical Cytology Grades and Cervical CancerREVIEWER COMMENTS

Reviewer #1 (Remarks to the Author): expert in AI pathology

Review comments

The authors have described an AI-based solution for cervical cancer diagnosis and grading, encompassing retrospective, prospective, and randomly sampled data from a substantial patient pool. Their methodology integrates patch level identification and slide classification, and the data presented demonstrates the efficacy of their method. However, prior to publication, I have several concerns:

Paper Presentation

1. The introduction provides a cursory background without delving into a systematic review of recent AI papers in the realm of cervical cancer screening. While it's commendable to mention AI applications in other diseases like prostate and breast cancer, the focus should remain on the advancements in cervical cytology screening. A few recent publications on this topic should be cited, discussed, and contrasted. Furthermore, parts of the discussion section would be better suited to the introduction. I will provide further comments on this below.
2. The organization and display of data in the results section are poor. Approximately 30-40% of the section seems to be a mere repetition of numerical values already present in the tables, making the paper cumbersome to follow, read, and interpret. I strongly advise the authors to invest more effort in data organization and consider graphical representations for clearer communication.
3. The database used in this study is only briefly outlined. It remains unclear whether the collected data is truly representative. A more detailed description of the database is needed.

Major Concerns

1. Data Consistency: There are discrepancies in the numbers presented, leading to confusion. I've elaborated on this in the minor comments.
2. Does the model's performance vary based on different sample preparation methods? We understand that different hospitals have their preparation protocols. Could this introduce performance discrepancies? The three mainly used solutions?
3. Paper Organization: For example, some parts of the discussion are better placed in the introduction. Refer to my annotations in the provided PDF.

4. The results section, laden with statistical data, could benefit from more concise and overarching statements in the "Conclusion and Discussion" section. The study's main outcomes need clearer highlighting.
5. The methodology for quality control of samples/images, be it manual or automated, isn't clearly stated. This is a pivotal step and requires elucidation.
6. Were there any observable variations between the two scanners utilized? If any.
7. Generalization and Data Augmentation: The section on color/data augmentation is vaguely explained. Generalization remains a perennial challenge in this field. It would be beneficial to understand any technical approaches used here to address color discrepancies rather than relying on random or blinded color augmentation.
8. Consider visually representing the results from Table 01 and appending the data sheet in the supplementary section for clarity.
9. Typically, HPV testing is either parallel or sequential for early cervical cancer diagnosis. It's unclear if this study also considered whether all patients undergoing cytological screening tested positive for HPV. What percentage of patients were HPV positive, if this was a factor at all?

Minor Issues

1. Line 60 repeats "HPV testing."
2. Line 62: Claiming it's challenging to establish a standardized system for population-based screening is misleading, as TBS is a widely accepted standard.
3. Line 70: Computer vision is distinct from machine learning, not a subfield.
4. Line 77's assertion about the extensive utilization of AI in pathological image analysis isn't wholly accurate. Despite AI's growing presence in cervical cancer cytology screening, its widespread use in pathology remains limited.
5. Line 85 contains inconsistent figures. For instance, Fig S1 cites 13,164, whereas another section mentions 16,056. This inconsistency is observed elsewhere too.
6. Line 164 indicates significant results regarding speed, but specific times are absent from the text, figures, tables, and supplementary data.
7. Line 190-195 seems misplaced. The relevance of this paragraph in its current section is questionable. Additionally, the styling and presentation of Figure 04 appear atypical.
8. Lines 203-233, discussing the current challenges in China and reviewing a recent publication, would be more appropriately placed in the introduction rather than the conclusion and discussion.
9. For additional minor comments, refer to my annotations in the provided PDF.

Reviewer #2 (Remarks to the Author): expertise in cervical cancer cytopathology

The authors describe a reproducible method in order to aid in the classification of cervical cytology. This is a technique that is currently being trialled not only in cervical cytology but also with the use of machine learning in order to aid in the evaluation of colposcopy images as well as in the Cytosponge Oesophageal cancer screening technique.

This of importance today as there is a dearth of qualified cytopathologists not only in China (as alluded to) but also in the UK and other HINC. Moreover, cytology training takes place even less now due to less cytology samples being generated in the UK due to hrHPV screening. Therefore this technology would have merit especially where cytopathologists are trying to improve their workflow. What is a strength of this paper is the fact that there are 3 cohorts - AI alone; AI + cytopathologist; cytopathologist alone. This gives a meaningful comparison and as expected the combination leads to a better diagnostic outcome.

While this paper is certainly not the first to demonstrate the efficacy of machine learning in this context. There are some significant differences that strengthen it in comparison to the current literature. First they had training on nearly 10000 images (whole slide) as well as tiles. This was then validated on not only an internal group but also an external group. This has two advantages - one it should reduce the overfitting experienced but more important would also be validated on patients of a different demographics but also with more sampling variation (which is a key factor in cytology). Secondly this paper also utilises a blinded approach to that of final validation in the cytopathologist vs. AI vs. cytopathologist and AI. This is crucial to demonstrating validity.

The weaknesses of the paper mainly stem from the following:

- 1) The patch level annotation of the cytology slides can be explained better; to a non-machine learning reader this is not clear and why such training is also important
- 2) I do not think the OR is the best statistical representation in comparing groups - I think the p value should suffice (though a statistical opinion should be considered for this) and in fact one other column would be to have AICS-assisted vs. cytopathologist
- 3) Negative predictive values should be tabulated as well

Overall I think the paper should be considered for publication with the above minor corrections.

Reviewer #3 (Remarks to the Author): clinical expertise in cervical cancer

This manuscript presents the development and validation of an Artificial Intelligence Cervical Cancer Screening (AICS) system designed for diagnosing cervical cytology grades and cancer. The study utilizes diverse datasets, including retrospective, prospective, and randomized controlled trial data, with a total of 16,056 participants. The performance of their analysis is assessed using the area under the curve (AUC) evaluation of receiver operating characteristic (ROC) curve analysis, revealing a notably high level of accuracy and efficiency in cervical cancer screening. The paper is well written, and the statistics is accurate. However, few comments are due:

1. My assessment of the authors' methodology is that it was appropriate for the study. However, I found it necessary to seek out additional details regarding their methodology. For pathologists and other readers of Nature Communications who are interested in potentially exploring the techniques described in the manuscript, it would be beneficial to present the methodological specifics in greater detail. For example, it would be helpful to elaborate on how the slides were annotated, whether individual cells or cell clusters were marked. The article mentions that negative cells do not require annotation, but it would be valuable to clarify whether any interference information, such as from infections like Chlamydia, fungi, or bacteria, present in the images, needs to be annotated.

2. As described in Figure 1 (Line 536), Squamous cell carcinoma (SCC) is a major diagnosis in cervical cytology. If clear squamous carcinoma cells can be observed, cytopathologists typically make a diagnosis of SCC. However, in the classification presented in Figure 2 (Line 550), there is no mention of SCC, and the text does not contain any information related to SCC. Please explain this discrepancy.

3. Intelligent cervical cytology detection has been the subject of several articles, some of which are also cited in this manuscript. Certain companies have even translated this research into products for clinical use. It would be beneficial to provide more specific references to the software, hardware, or resources used for data analysis within the AICS deep learning framework. In the "Discussion" section, please provide a more detailed explanation of the innovations presented in this article.

4. More and more research suggests that intelligent analysis is not merely a "black box" effect. Many "pathological features" or "clues" can be recognized by machines through analysis. As readers of Nature Communications, many people might be interested in how the computer makes these judgments. Have the authors attempted retrospective analysis to determine precisely which pathological features the computer recognizes to make its judgments? Is this in line with what cytologists observe morphologically? This would be a fascinating aspect to explore.

RESPONSE TO REVIEWERS' COMMENTS

Reviewer #1 (Remarks to the Author): expert in AI pathology

Review comments

The authors have described an AI-based solution for cervical cancer diagnosis and grading, encompassing retrospective, prospective, and randomly sampled data from a substantial patient pool. Their methodology integrates patch level identification and slide classification, and the data presented demonstrates the efficacy of their method. However, prior to publication, I have several concerns:

Paper Presentation

1. The introduction provides a cursory background without delving into a systematic review of recent AI papers in the realm of cervical cancer screening. While it's commendable to mention AI applications in other diseases like prostate and breast cancer, the focus should remain on the advancements in cervical cytology screening. A few recent publications on this topic should be cited, discussed, and contrasted. Furthermore, parts of the discussion section would be better suited to the introduction. I will provide further comments on this below.

Response: Thank you for your feedback and your detailed comments on the introduction section of the paper. I appreciate your input and agree with your observations. I have added a more in-depth background that includes a systematic review of recent AI papers specifically related to cervical cancer screening, please see the revised version. This will help establish the context and significance of the topic. Your feedback is highly valuable in refining this research. Thank you once again for your insights.

2. The organization and display of data in the results section are poor. Approximately 30-40% of the section seems to be a mere repetition of numerical values already present in the tables, making the paper cumbersome to follow, read, and interpret. I strongly advise the authors to invest more effort in data

organization and consider graphical representations for clearer communication.

Response: Thank you for your kind remind. We are sorry for the poor organization in this manuscript. In accordance to your suggestion, we have some modification the data organization throughout the whole manuscript, please see the revised version.

3. The database used in this study is only briefly outlined. It remains unclear whether the collected data is truly representative. A more detailed description of the database is needed.

Response: Thanks for your kind suggestion. As we mentioned in the manuscript (line 304-306, page 16), the participant eligibility assessment and strict specimen criteria: Participants, 18 years or older, not pregnant, without mental illness or cognitive impairment, and who consented to cervical liquid-based cytology for definite diagnosis, were included. Satisfactory specimens required at least 5,000 visible and uncovered squamous epithelial cells and abnormal cells (atypical squamous cells or atypical glandular cells and above). Exclusion criteria were participants who had undergone cervical resection with primary malignancies. Unsatisfactory specimens were identified by fewer than 5,000 visible, uncovered squamous epithelial cells, or more than 75% of squamous epithelial cells affected due to blood, inflammatory cells, epithelial cell overlapping, poor fixation, excessive drying, or contamination of unknown components, as defined in the 2014 Bethesda System for Reporting Cervical Cytology (TBS 2014). Therefore, the participants and specimen collected in this study is representative. However, as the baseline information including age, race, etc. were not recorded, because this is a sex-specific study and this study was a screening program in China, age had no effect on the results, so we didn't count the age distribution, which is a limitation in this study. Thank you once again for your insights.

Major Concerns

1. Data Consistency: There are discrepancies in the numbers presented, leading to confusion. I've elaborated on this in the minor comments.

Response: Thanks for your kind suggestion. We have revised throughout the

manuscript in accordance to your suggestion. Thank you once again for your insights.

2. Does the model's performance vary based on different sample preparation methods? We understand that different hospitals have their preparation protocols. Could this introduce performance discrepancies? The three mainly used solutions?

Response: Thank you for your constructive questions. The performance of AICS may indeed exhibit variations based on differing sample preparation methods, a notion we acknowledge. It is recognized that different hospitals adhere to their distinct preparation protocols. These variations in sample preparation can manifest as performance discrepancies, as illustrated in and Figure 3A, wherein the model exhibits superior performance on the internal dataset as compared to the other two external datasets.

However, it is imperative to highlight that several measures have been undertaken during the training phase to attenuate these discrepancies. Such measures encompass both conventional image data augmentation and stain augmentation techniques, as briefly delineated in lines 361-366, page 18 .

Specifically, stain augmentation was accomplished by executing color deconvolution on patch images to acquire a 3x3 stain component matrix. This stain component matrix was subsequently subjected to random perturbations to emulate varying staining intensities throughout the training process.

Thank you once again for your insights.

3. Paper Organization: For example, some parts of the discussion are better placed in the introduction. Refer to my annotations in the provided PDF.

Response: Thank you for advise on the paper organization. According to your advice, some parts of the discussion have been placed in the introduction section (lines 79-91, page 5): “The development of an AI-based cervical cancer screening system has the potential to revolutionize the field of cervical cancer diagnosis. Traditional methods of cervical cytology grading rely on manual examination by pathologists, which can be

time-consuming, subjective, and prone to inter-observer variability. While cervical cytology is a widely accepted examination method, it heavily relies on the expertise and experience of cytopathologists. Microscope readings suffer from poor reproducibility and are susceptible to various interfering factors. In contrast, AI screening has the potential to enhance the consistency of cytopathological results, improve the consistency of comparison with biopsy results, increase sensitivity, and reduce the risk of misdiagnosis. In a previous study, a comprehensive cervical liquid-based cytology smear TBS diagnostic system (AIATBS system) was developed. It employed YOLOv3 for target detection, Xception and patch-based models for target classification, and U-net for nucleus segmentation. This AI system trained models for 24 classifications. However, these characteristics posed challenges for the broad practical application of the AI model and increased the complexity of screening tasks.” Thank you once again for your insights.

4. The results section, laden with statistical data, could benefit from more concise and overarching statements in the “Conclusion and Discussion” section. The study's main outcomes need clearer highlighting.

Response: Thank you for advise on data presentation. Thank you for your feedback, especially regarding the presentation of the results and their integration into the "Conclusion and Discussion" section. I appreciate your input and agree that the balance between detailed statistical data in the results section and more concise, overarching statements in the conclusion and discussion is crucial for the readability and comprehension of the paper. Your feedback is valuable in ensuring the paper's clarity and effectiveness in conveying the research findings. I have made the necessary adjustments to enhance the overall presentation of the study's outcomes (lines 214-229, pages 11-12): “This study aimed to develop and validate an AICS for the diagnosis of cervical cytology grading and cervical cancer through the analysis of whole slide images of cervical cells. AICS was trained and tested on diverse datasets with 16,056 participants. It utilized two AI models: cell detection and whole slide image classification. The research utilized a multicenter, retrospective, and prospective

population dataset, along with a randomized controlled trial to validate the system. Through the validation of these datasets, AICS demonstrated performance in identifying cervical cytology grading. In prospective evaluation, it reached an AUC of 0.947, sensitivity of 0.946, specificity of 0.890, and accuracy of 0.892. Notably, in a randomized controlled trial, AICS-assisted cytopathologists outperformed, with significantly higher AUC, specificity, and accuracy, and a remarkable 6.2% improvement in sensitivity. AICS shows promise as an adjunct tool for precise and efficient cervical cancer screening. The results of this study showed promising outcomes for AICS in accurately identifying cervical cytology grading. The system achieved a high sensitivity and specificity, demonstrating its ability to detect abnormal cells and differentiate them from normal cells effectively. This could potentially lead to earlier detection and intervention for cervical cancer, improving patient outcomes and reducing the burden on healthcare systems. ” Thank you for your constructive input.

5. The methodology for quality control of samples/images, be it manual or automated, isn't clearly stated. This is a pivotal step and requires elucidation.

Response: Thank you for your insightful suggestions on the quality control measures. Firstly, in the methods section titled "Whole Slide Image Acquisition and Preprocessing", the inclusion and exclusion criteria for patients and smears was elucidate (lines 307-309, pages 16), which are diligently applied manually, ensuring the data quality within our study.

Furthermore, alongside these manual quality control measures, we have instituted an AI-assisted approach to discern and address potential scanning quality issues during the digitization process. To facilitate this, we developed an image classification model utilizing thumbnail images of Whole Slide Images (WSIs) to detect instances of scanning quality hindrances, such as blurriness or incomplete scanning areas. The thumbnails' short edge was standardized to 1000 pixels, representing scaled-down versions of the original WSIs. Images pinpointed by this model as potentially problematic were subjected to a subsequent manual review, and upon confirmation of scanning quality issues, were excluded from our dataset.

The quality model is architected around three central components: the EfficientNet Backbone, a quality summary branch, and a quality detail branch. We employ EfficientNet for model backbone purposes, particularly in feature extraction. The quality summary branch is tasked with rendering an overall assessment of slide quality, performing binary classification to categorize slides as either acceptable or problematic. In addition, the quality detail branch offers a more nuanced evaluation by estimating the severity of specific quality issues, employing a multi-label classification technique to identify and categorize different types of quality problems within the slides.

To sum up, the model acts as a preliminary filter, with WSIs identified as potentially problematic being subjected to a secondary manual review. WSIs confirmed to exhibit scanning quality issues were omitted from our datasets. In addition to its role in dataset preparation, this quality control model is instrumental in the practical application of AIC. It aids in identifying potentially problematic slides, thereby bolstering the system's overall reliability and assisting cytopathologists in their daily diagnostic endeavors.

It warrants mention that this quality module operates as an independent component within our in-house developed deep learning framework, rendering it adaptable to any quality process within machine learning products or projects. Thank you once again for your insights.

6. Were there any observable variations between the two scanners utilized? If any.

Response: Thank you for your constructive suggestions, we are sorry for not describing the variations between two scanners in the manuscript. In fact, we have done a validation in the internal validation dataset in the preliminary preparation stage, the internal validation dataset encompasses a total of 2,152 Whole Slide Images (WSIs), which includes 1,337 WSIs from the UNIC scanner and 815 WSIs from the KFB scanner, and the scanning quality between the two scanners demonstrates subtle variations in visually discernible outcomes. The performance of AICS on WSIs from both scanners is elucidated in the table below. Thank you once again for your insights.

Table 1 The performance of different scanners in internal validation dataset.

	KFB	UNIC	Total
Sensitivity(95%CI)	0.926 (0.908-0.944)	0.889 (0.872-0.906)	0.906 (0.894-0.918)
Specificity(95%CI)	0.779 (0.751-0.808)	0.927 (0.913-0.941)	0.874 (0.860-0.888)
Balanced Accuracy(95%CI)	0.853 (0.838-0.868)	0.908 (0.920-0.920)	0.890 (0.877-0.903)
AUC(95%CI)	0.879 (0.856-0.901)	0.949 (0.961-0.961)	0.922 (0.911-0.933)

7. Generalization and Data Augmentation: The section on color/data augmentation is vaguely explained. Generalization remains a perennial challenge in this field. It would be beneficial to understand any technical approaches used here to address color discrepancies rather than relying on random or blinded color augmentation.

Response: We are grateful for the suggestion. The color augmentation and generalization should be explained clearly in the methods section to address color discrepancies.

During the training process, color augmentation was applied systematically rather than blindly or randomly. A specialized staining augmentation technique was utilized as delineated by Macenko et al. (Reference: Macenko M, Niethammer M, Marron JS, et al. A method for normalizing histology slides for quantitative analysis. 2009 IEEE International Symposium on Biomedical Imaging: From Nano to Macro. IEEE, 2009: 1107-1110.), which is particularly pertinent to pathological image processing. This technique aims to alleviate potential performance discrepancies arising from staining variations.

The specific steps encompassed within this augmentation procedure are outlined as follows: Initially, a given RGB image was transformed into the Stain Density

Absorbance (SDA) space. Subsequently, the Macenko method was employed to perform color deconvolution, yielding a 3x3 stain component matrix. Each column of this matrix represents a distinct stain component, such as Hematoxylin (H) and Eosin (E). Random perturbations were then introduced column-wise to the 3x3 stain component matrix. Finally, the perturbed stain component matrix was utilized to reconstruct the SDA image back into the RGB space.

This approach ensured that the color augmentation process was both controlled and consistent with prevalent practices in pathological image processing, thereby mitigating potential performance variations induced by staining differences.

Example for color augmentation:

1. Jitter on the E column

2. Jitter on the H column

3. Jitter on both H and E columns

In accordance with the reviewer concerns, we have added a brief description as follows (lines 361-366, pages18-19): “Data augmentation especially color augmentation was applied systematically rather than blindly or randomly, which plays a critical role in enhancing the accuracy of deep learning object detection frameworks and mitigating overfitting. The specific steps encompassed within this augmentation procedure are outlined as follows: Initially, a given RGB image was transformed into the Stain Density Absorbance (SDA) space. Subsequently, the Macenko method was employed to perform color deconvolution, yielding a 3x3 stain component matrix.” Thank you once again for your insights.

8. Consider visually representing the results from Table 01 and appending the data sheet in the supplementary section for clarity.

Response: Thank you for advise on data representation. In according to the work suggested by the reviewer, we had presented different subgroups’ AUCs curves of retrospective dataset, prospective dataset, and randomized controlled trial in Figure 4, Figure 5, and Figure 6 respectively in the revised manuscript. Thank you once again for your insights.

9. Typically, HPV testing is either parallel or sequential for early cervical cancer diagnosis. It's unclear if this study also considered whether all patients undergoing

cytological screening tested positive for HPV. What percentage of patients were HPV positive, if this was a factor at all?

Response: Thank you for your kind suggestion. Currently, there are some guidelines on cervical cancer screening worldwide. Among the most notable ones are the guidelines from the American Society of Clinical Oncology (ASC), the American Society for Colposcopy and Cervical Pathology (ASCCP), and the United States Preventive Services Task Force (USPSTF). Depending on the specific population for screening, different screening recommendations are given:

2012 ASCCP guideline: female aged from 21 to 29 recommend doing cytology screening every 3 years and aged from 30 to 65: recommend having cytology screening every 3 years or co-testing every 5 years.

2018 USPSTF guideline: female aged from 21 to 29 recommend having cytology screening every 3 years and aged 30-65 recommend having cytology screening every 3 years, or HPV testing every 5 years, or co-testing every 5 years.

2020 ACS guideline: female aged from 25 to 65, if primary HPV testing is not accessible, it's recommended to have cytology screening annually or HPV co-testing every 5 years.

Therefore, HPV testing is either parallel or sequential for early cervical cancer screening. In this study, we didn't track the result whether participants underwent HPV testing with positive cytopathological results. Therefore, we are unable to provide the information about percentage of patients who are HPV positive. As the main study outcome in this study is aimed to train and validate an Artificial Intelligence Cervical Cancer Screening (AICS) system for diagnosing cervical cytology grades, the lack of HPV testing result won't affect the performance of AICS. Thank you once again for your insights.

Minor Issues

1. Line 60 repeats "HPV testing."

Response: Thank you for your kind suggestion. We have modified the description

accordingly (lines 58-60, page 4): “Currently, cervical cancer screening methods mainly include cervical cytology screening, HPV testing, and DNA ploidy testing.”

2. Line 62: Claiming it's challenging to establish a standardized system for population-based screening is misleading, as TBS is a widely accepted standard.

Response: Thank you for your kind remind. We have modified the sentence according to the your suggestion (lines 61-63, page 4): “However, there is a significant shortage of cytopathologists worldwide and the lack of qualified and experienced cytopathologists in low-resource areas has resulted in a false negative rate of over 10% in routine diagnosis results.”.

3. Line 70: Computer vision is distinct from machine learning, not a subfield.

Response: Thank you for your kind suggestion. We have modified the sentence according to the your suggestion (lines 72-73, page): “ Computer vision has been widely utilized in disease detection.”.

4. Line 77's assertion about the extensive utilization of AI in pathological image analysis isn't wholly accurate. Despite AI's growing presence in cervical cancer cytology screening, its widespread use in pathology remains limited.

Response: Thank you for your insightful comments. We have modified the sentence according to the your suggestion (lines 75-78, pages 4-5): “”Promising results have also been achieved in breast cancer detection, with several studies applying deep learning techniques to mammography and digital breast tomosynthesis classification, showcasing excellent performance when evaluated with extensive datasets.

5. Line 85 contains inconsistent figures. For instance, Fig S1 cites 13,164, whereas

another section mentions 16,056. This inconsistency is observed elsewhere too.

Response: Thank you for your kind suggestion. As we shown in the Fig S1, there are 13,164 participants screened for eligibility at Sun Yat-sen memorial Hospital (SYSMH), and finally 11,468 participants meet the inclusion criteria. In addition, there are 600 participates from Guangzhou Women and Children Medical Center (GWCMC) and The Third Affiliated Hospital of Guangzhou Medical University (TAHGMU) each. The SYSMH prospective dataset included 2,780 participants, and the randomized controlled trial enrolled 680 participants. Therefore, there are 16,056 participants meet the inclusion criteria and quality control. Thank you once again for your insights.

6. Line 164 indicates significant results regarding speed, but specific times are absent from the text, figures, tables, and supplementary data.

Response: Thank you for your kind suggestion, and we are sorry for not state the speed in diagnose. To be more clear and in accordance with the reviewer concerns, we have added a brief description in lines 208-209, page 11 : “The time required for AICS analysis of a whole-slide image (WSI) is within 120 seconds.” Thank you once again for your insights.

7. Line 190-195 seems misplaced. The relevance of this paragraph in its current section is questionable. Additionally, the styling and presentation of Figure 04 appear atypical.

Response:

Reponse: Thank you for your kind remind. According to your suggestion, we have revised the description to “For participants with ASC-US+, AUCs were 0.944 in the AICS alone, 0.959 in cytopathologist, and 0.995 in AICS-assisted groups (Figure 6B). LSIL+ and HSIL participants achieved high AUCs over 0.980 (Figure 6C-6D).”, see lines 201-204, page 11.

The Figure4 in the original manuscript is presented the working mode of AICS. AICS has been utilized in over 30 hospitals in China. For patients who coming to a collaborating hospitals for thinprep cytologic test, the platform is able to provide two

key clinical applications: 1) get the precision diagnosis of cervical cytology grades with the assistance of AICS, while the cervical cytology smears upload to the AICS, so as to increase the accuracy of Diagnosis; and 2) The AICS provides free access as a consulting service for patients and clinicians after their have uploaded their WSI on the AICS. Experienced experts will discuss the complex case and arrive at a consensus diagnosis. Thank you once again for your insights.

8. Lines 203-233, discussing the current challenges in China and reviewing a recent publication, would be more appropriately placed in the introduction rather than the conclusion and discussion.

Response: Thank you for your feedback and your specific suggestion regarding lines 203-233 of the paper. I appreciate your input. I have made the necessary adjustments to relocate this content to the introduction section, ensuring a more appropriate structure for the paper (lines 64-66, page 4): “Currently, China has a relatively low overall cervical cancer screening rate compared to developed countries, primarily due to a shortage of pathologists. In 2013, there were less than 20,000 registered pathologists in China, while the actual demand was nearly 100,000.”.

Your feedback is valuable in improving the overall organization and presentation of the research. Thank you for your insights.

9. For additional minor comments, refer to my annotations in the provided PDF.

Response: Thank you for advise on the paper revision. We have modified the manuscript throughly, please see the revised version manuscript. Thank you for your insights.

Reviewer #2 (Remarks to the Author): expertise in cervical cancer cytopathology

The authors describe a reproducible method in order to aid in the classification of cervical cytology. This is a technique that is currently being trialled not only in cervical cytology but also with the use of machine learning in order to aid in the

evaluation of colposcopy images as well as in the Cytosponge Oesophageal cancer screening technique.

This is of importance today as there is a dearth of qualified cytopathologists not only in China (as alluded to) but also in the UK and other HINC. Moreover, cytology training takes place even less now due to less cytology samples being generated in the UK due to hrHPV screening. Therefore this technology would have merit especially where cytopathologists are trying to improve their workflow. What is a strength of this paper is the fact that there are 3 cohorts - AI alone; AI + cytopathologist; cytopathologist alone. This gives a meaningful comparison and as expected the combination leads to a better diagnostic outcome.

While this paper is certainly not the first to demonstrate the efficacy of machine learning in this context. There are some significant differences that strengthen it in comparison to the current literature. First they had training on nearly 10000 images (whole slide) as well as tiles. This was then validated on not only an internal group but also an external group. This has two advantages - one it should reduce the overfitting experienced but more important would also be validated on patients of a different demographics but also with more sampling variation (which is a key factor in cytology). Secondly this paper also utilises a blinded approach to that of final validation in the cytopathologist vs. AI vs. cytopathologist and AI. This is crucial to demonstrating validity.

The weaknesses of the paper mainly stem from the following:

1) The patch level annotation of the cytology slides can be explained better; to a non-machine learning reader this is not clear and why such training is also important

Response: We really appreciate your insightful comments, and can't agree with you more than the patch level annotation of the WSIs should be elucidated in the method section.

The annotation of Whole Slide Images (WSIs) at the patch level entails the segmentation of high-resolution WSIs into smaller patches, followed by the labeling of individual abnormal cells within a selected subset of these patches. During this meticulous process, experienced cytopathologists delineate bounding boxes around these abnormal cells and specify their respective types. These patch-level annotations serve as a foundational dataset for training the cell-level deep neural network detection model.

In this study, patch-level annotation of WSIs encompasses two distinct phases: an initial manual annotation phase and an AI-suggested annotation phase. In the initial manual annotation phase, cytopathologists engage in labeling a subset of patches, predominantly those embodying cells deemed highly representative or typical. Subsequent to the initial phase of manual annotation, a detection model is trained and then deployed to perform sliding window inference on WSIs, thereby generating AI-recommended Regions of Interest (ROIs), encompassing cells identified as positive. These AI-recommended ROIs undergo review and annotation by cytopathologists prior to their integration into our patch-level training dataset. This iterative procedure, which involves the confirmation and amendment of AI-recommended ROIs by cytopathologists, ensures a progressive refinement of AI performance based on expert cytopathological input. This methodology facilitated the construction of our training dataset with high-quality annotations while expediting the annotation process.

Therefore, pertaining to the training process of the cell-level deep neural network detection model, it holds significance as it educates the model to discern abnormal cells within WSIs, each of which may harbor tens of thousands of cells. The output generated by the cell detection model serves as the input for the WSI-level classification models, underscoring its crucial role in the overarching analytical framework.

To be more clear and in accordance with the reviewer concerns, we have added a brief description in the method section. Thank you for your insights.

2) I do not think the OR is the best statistical representation in comparing groups - I think the p value should suffice (though a statistical opinion should be

considered for this) and in fact one other column would be to have AICS-assisted vs. cytopathologist

Response: Thank you for your insightful advice. We do agree that P value is sufficient for statistical representation in comparison, therefore we have modified the Table 2 and Table 3 in the manuscript. In addition, according to your kind remind, we have add the presentation of AICS-assisted vs. Cytopathologists in the performance of AICS, cytopathologists and AICS-assisted in the SYSMH prospective validation datasets (see Table2). Thank you once again for your insights.

3) Negative predictive values should be tabulated as well

Response: Thank you for your insightful suggestion. We have added the NPV for retrospective and prospective validation datasets and randomized controlled trial in Table S5 and Table S6 in Supplement.

The NPV is 0.980 in SYSMH, 0.912 in GWCMC and 0.963 in TAHGMU in the validation datasets. The NPV is 0.997 in AICS alone, 0.996 in cytopathologist and 0.996 in AICS-assisted in the SYSMH prospective validation datasets. The NPV is 0.994 in AICS alone, 0.989 in cytopathologist and 1.000 in AICS-assisted in the randomized controlled trial. Thank you for your insights.

Reviewer #3 (Remarks to the Author): clinical expertise in cervical cancer

This manuscript presents the development and validation of an Artificial Intelligence Cervical Cancer Screening (AICS) system designed for diagnosing cervical cytology grades and cancer. The study utilizes diverse datasets, including retrospective, prospective, and randomized controlled trial data, with a total of 16,056 participants. The performance of their analysis is assessed using the area under the curve (AUC) evaluation of receiver operating characteristic (ROC) curve analysis, revealing a notably high level of accuracy and efficiency in cervical cancer screening. The paper is well written, and the statistics is accurate. However, few comments are due:

1. My assessment of the authors' methodology is that it was appropriate for the study. However, I found it necessary to seek out additional details regarding their methodology. For pathologists and other readers of Nature Communications who are interested in potentially exploring the techniques described in the manuscript, it would be beneficial to present the methodological specifics in greater detail. For example, it would be helpful to elaborate on how the slides were annotated, whether individual cells or cell clusters were marked. The article mentions that negative cells do not require annotation, but it would be valuable to clarify whether any interference information, such as from infections like Chlamydia, fungi, or bacteria, present in the images, needs to be annotated.

Response: Thank you for your kind remind. We really should elucidate the method section more specifically, especially on annotation section.

The patch-level annotation of Whole Slide Images (WSIs) involves dissecting high-resolution WSIs into smaller patches and labeling individual abnormal cells and cell clusters within a selected subset of these patches. Throughout this process, experienced cytopathologists delineate bounding boxes around these abnormal cells, specifying their types. These patch-level annotations are subsequently utilized to train the cell-level deep neural network detection model.

In this study, patch-level annotation of WSIs encompasses two distinct phases: an initial manual annotation phase and an AI-suggested annotation phase. In the initial manual annotation phase, cytopathologists engage in labeling a subset of patches, predominantly those embodying cells deemed highly representative or typical. Subsequent to the initial phase of manual annotation, a detection model is trained and then deployed to perform sliding window inference on WSIs, thereby generating AI-recommended Regions of Interest (ROIs), encompassing cells identified as positive. These AI-recommended ROIs undergo review and annotation by cytopathologists prior to their integration into our patch-level training dataset. This iterative procedure, which involves the confirmation and amendment of AI-recommended ROIs by cytopathologists, ensures a progressive refinement of AI performance based on expert

cytopathological input. This methodology facilitated the construction of our training dataset with high-quality annotations while expediting the annotation process.

Besides, as mentioned in the manuscript, negative smears do not necessitate annotation. This is predicated on our patch-level annotation approach, which marks all abnormal cells within the individual patches. Consequently, any unannotated regions within the patches are construed as background during the training of our detection model. This encompasses patches where no abnormal cells were identified within the AI-suggested ROIs during the AI-suggested annotation round, enabling the model to effectively distinguish abnormal cells from the background.

With respect to the presence of interference information, such as infections like Chlamydia, fungi, or bacteria in the images, we acknowledge their presence. However, these elements were not manually annotated in our dataset. Although not explicitly annotated by pathologists, the presence of such elements in the background was nonetheless considered during our model training process. These elements were treated as part of the negative background, empowering the model to differentiate them from the abnormal cells. Thank you for your insights.

2. As described in Figure 1 (Line 536), Squamous cell carcinoma (SCC) is a major diagnosis in cervical cytology. If clear squamous carcinoma cells can be observed, cytopathologists typically make a diagnosis of SCC. However, in the classification presented in Figure 2 (Line 550), there is no mention of SCC, and the text does not contain any information related to SCC. Please explain this discrepancy.

Response: Thank you for your kind remind. As you mentioned, squamous cell carcinoma (SCC) indeed holds a significant diagnostic value in this context and is critical in clinical practice. In accordance with your suggestions, the detailed information regarding samples of SCC in this study was shown in Table S1. There are 35 (0.4%), 5 (0.2%) , 4 (0.7%) and 2 (0.1%) cases in SYSMH training dataset, SYSMH internal validation dataset, GWCMC external validation dataset, and SYSMH prospective validation dataset, respectively. And there no SCC sample in TAHGMU External Validation Dataset and randomized controlled trial.

As depicted in Figure 1, SCC were distinctly labeled due to their diagnostic significance. However, as illustrated in Figure 2, SCC is not explicitly delineated as a separate WSI (Whole Slide Image) category. This is because, for the scope of our study, we amalgamated SCC with HSIL (High-grade Squamous Intraepithelial Lesion) as a category named HSIL+.

This decision was undergirded by several salient reasons:

1. Morphology: In numerous instances, the cytological morphology of SCC cells exhibits overlap with that of HSIL cells. The demarcation between these two categories based solely on cytopathological appearance can be challenging. Within the purview of an AI-assisted diagnostic system—aimed at furnishing reliable diagnostic references to support cytopathologists—merging categories with morphological overlap can attenuate ambiguity and provide a more harmonized reference.

2. Clinical Management: In accordance with the 2019 ASCCP (American Society for Colposcopy and Cervical Pathology) guidelines, the clinical management for both SCC and HSIL is identical, necessitating immediate intervention. This congruence in clinical management further substantiates our decision to merge these categories.

Thank you for your insights.

3. Intelligent cervical cytology detection has been the subject of several articles, some of which are also cited in this manuscript. Certain companies have even translated this research into products for clinical use. It would be beneficial to provide more specific references to the software, hardware, or resources used for data analysis within the AICS deep learning framework. In the "Discussion" section, please provide a more detailed explanation of the innovations presented in this article.

Response: Thanks for your constructive comments. We do really need to provide relevant information for data analysis within the AICS deep learning framework, and please find the detailed information below, see Table 2.

Table2. General parameters of hardware in AICS

	Description
Solid state drive	Intel S4610 480G SSD
Mechanical hard disk	WD8TB 7.2K 3.5-inch
Graphics card	RTX 2080Ti
Driver version	510.47.03

Also, we couldn't agree more that more details on the innovation should be elucidated in the manuscript. As mentioned in the manuscript: "The research utilized a multicenter, retrospective, and prospective population dataset, along with a randomized controlled trial to validate the system. Through the validation of these datasets, AICS demonstrated performance in identifying cervical cytology grading.". in addition, the development of the AICS system commenced in 2019 and underwent verification with a substantial volume of retrospective and prospective data by October 2020. At that juncture, we leveraged the most avant-garde deep learning technology for cervical cytology screening. Concurrently, to the best of our knowledge, there were no published studies mirroring our endeavor. The core innovation of this system lies in its optimal amalgamation of the latest deep learning technology and sophisticated domain knowledge, garnered from top-tier cytopathologists in China and real-world clinical data. Since then, our models and product have perpetually evolved, in tandem with the advancements in deep learning technology and the accrual of data, courtesy of the extensibility and adaptability engineered into our framework.

Our system is architected on a cutting-edge Docker-based micro-service infrastructure, wherein different functional modules are encapsulated as callable micro-services. Any subsequent AI model updates can be seamlessly integrated into the system and orchestrated via a configuration tool embedded within the AICS framework. The AICS system embodies a comprehensive pipeline designed to assist cytopathologists in conducting routine diagnoses and screening tasks for cervical cancer. Specifically, it encompasses an AI-assisted quality control module, domain-specific stain augmentation processes, iterative AI-assisted annotation procedures, cell-level cytological feature extraction, slide-level feature statistical distribution, and multi-scale

deep learning algorithms.

As of now, our latest AICS system has been deployed in over 30 hospitals across China, manifesting its real-world impact.

Thank you for your insights.

4. More and more research suggests that intelligent analysis is not merely a "black box" effect. Many "pathological features" or "clues" can be recognized by machines through analysis. As readers of Nature Communications, many people might be interested in how the computer makes these judgments. Have the authors attempted retrospective analysis to determine precisely which pathological features the computer recognizes to make its judgments? Is this in line with what cytologists observe morphologically? This would be a fascinating aspect to explore.

Response: We appreciate your time and effort in reviewing this manuscript, and we are grateful for your insightful question. Indeed, the interpretability of deep learning outputs is often challenged by domain experts, particularly in the medical domain. This remains a significant area of research within the field of deep learning. Many researchers resort to utilizing heat maps to visualize the potential correlation between the outputs of deep learning models and the morphological features of images. In this article, the model was achieved by implementing the D-RISE (Detector Randomized Input Sampling for Explanation) algorithm (<https://arxiv.org/abs/2006.03204>), which overlays a detection heat map onto the original image. D-RISE adapts the RISE technique (<https://arxiv.org/abs/1806.07421>) for object detection models, thereby presenting the detection bounding box and heat maps within the same area. Within the heat map, the intensity of color indicates the level of importance that a pixel holds to

the model output, enabling an analysis of which pixels are of greater significance to the detection model.

The figure illustrates six types of abnormal cells identified by our detection model, alongside corresponding heat maps. Evidently, our model is proficient in detecting these cells or cell clusters exhibiting varying grades of abnormality and sizes, in a manner largely congruent with the morphological characteristics utilized by cytopathologists. For instance, the detection of abnormal squamous epithelial cells is characterized by enlarged nuclei, a high nuclear-to-cytoplasmic ratio, and a perinuclear cavity, among others. Detailed descriptions of the morphological characteristics are provided below:

1. ASC-US: This category showcases an intermediate squamous cell featuring an enlarged nucleus and slight nuclear membrane irregularity.
2. LSIL: This delineates a cell with a sharply defined perinuclear cavity, condensation of cytoplasm around the periphery, and abnormal nuclear features including enlargement, binucleation, and nuclear membrane irregularity.
3. ASC-H: This features a group of atypical immature metaplastic cells with enlarged nuclei, a high nuclear-to-cytoplasmic ratio, coarse chromatin, and irregular nuclear contour. The cytologic attributes are indicative of concern, albeit insufficient for an HSIL interpretation.
4. HSIL: This category portrays crowded groups with variably sized cells, alongside nuclear envelope irregularities and abnormal chromatin.
5. SCC: Here, malignant cells exhibit variable shapes and sizes, with some showcasing keratinized "tadpole cells." The nuclei range from vesicular with irregular nuclear contours and nucleoli, to pyknotic in the keratinized cells.
6. AGC: This represents small groups of cells with slightly enlarged nuclei, small nucleoli, and vacuolated cytoplasm.

In accordance with your comments, we have added a description in the method section (lines 344-350, pages 18) in our revised manuscript and exhibited a few examples of detected abnormal cells (Figure S4 in revised manuscript). Thank you once again for your insights.

REVIEWER COMMENTS

Reviewer #1 (Remarks to the Author):

Review comments

The revised manuscript appears to have undergone only minor amendments. The overall improvement in the quality of the manuscript is modest, and there are still many areas that need careful attention and revision. Consistency in terms, figures, tables, and clarity of presentation are crucial for readers' comprehension.

Abbreviation Consistency:

- The abbreviation for the system name "Artificial Intelligence Cervical Cancer Screening" should be "AICCS" instead of "AICS." Please ensure the abbreviation is consistent throughout the manuscript.
- On line 431, the system is referred to as "Automated Interpretation of Cervical Smears." If this is the correct full name, "AICS" is appropriate. Kindly verify and maintain consistency.

Paper Presentation:

1. Figures 1 and 2 are densely packed with information. I recommend labelling different sections for clarity and elaborating on the figure captions.
2. The presentation of data in the results section needs revision. There is an unnecessary repetition of numerical values found in tables. It might be more effective to represent some of this data graphically. This lack of improvement was previously highlighted.
3. Ensure that content is placed appropriately in the designated sections. I've noted specific concerns on the provided PDF.
4. Enhance the resolution of the figures and tables, particularly Table S1, which is currently too blurry to decipher.
5. The manuscript's language quality requires considerable refinement to avoid misunderstanding and enhance readability.

Major Concerns

1. Clearly define classifications like LSIL+ and ASCUS+ early in the manuscript and use them consistently.
2. The classification of samples appears inconsistent. Clarify the criteria for including SCC in certain classifications.
3. The data referenced on lines 162-167 isn't located in Table S4. Please clarify its source.
4. Ensure consistency between the main text and tables. I've highlighted a few discrepancies, but a thorough review is required.
5. Figure 5 and 6 are identical? Please confirm. According to the text line 104-105, the prospective study included 2780 samples and 680 samples for a randomized controlled trial sample. In the case the system is robust, it is possible that the overall accuracy is very close, while I wonder if the AUC curves are identical. From the Figure 5 and 6, the AUC curves look identical. Please check if this is a mistake.
6. The system included patch-level detection and classification and whole slide classification. The patch-level results are not reported, which is actually important to understand the work.
7. At the cell level, as morphology is one of the critical factors, probably the dominant feature, such details and results are missing in the work.
8. The anonymization of patient images shouldn't be a hindrance to data public availability.

Minor Issues

1. Line 57. Detecting ... and halting... are (not "is") crucial. There are some other grammar mistakes and improper expressions, which are difficult for me to correct all. Minor language problems are not big issues making the reading difficult and sometimes misleading. Strongly suggest the authors to check the language.
2. Line 65, the data in 2013 is out of date, which is about 10 years ago.
3. Line 92, please define the abbreviation according to the spelling of the full name.
4. Line 116-117, what does this sentence mean? See my remark in the PDF file.
5. Not sure why this information is included in Supp. An overview of the data distribution is one of the key aspects of AI paper. I was also asking about how representative the data is.
6. Line 30, I would only guess data in line 130-134 are from some tables. And I found it in Table 1.
7. Line 142. Please make sure that the numbers reported in the main text are the same as in the Tables. I checked some of them and tried to understand the paper better, but I did not check all. Please make sure there is no copy-paste mistake and the numbers are consistent.
8. Line 148 – 154, please make sure, the reported number is not misleading. The diagnostic accuracy is reported first and then without any explanation, the sensitivities are reported. Without checking the table, it is important to understand that the following numbers are sensitivities.

9. Line 162 – 166, such data is not really reported in Table S4. I failed to locate such information.
10. Line 180 – 191. Please make sure that the numbers reported are correct and accurate.
11. Line 213, language.
12. Line 233, The adaptiveness of AICS is not clearly demonstrated in the work and this is an overclaim.
13. Line 234-236. The comparison of the AICS with AIATBS might not be an apple-to-apple comparison as the testing databases are different.
14. Line 238, there might be some system designed for future adaptive learning, while the paper itself does not yet provide the evidence to support the claim of the adaptive capability of AISC.
15. Line 243-245, if follows TBS, there are more than 5 classes.
16. Line 246. A few Class II solutions were approved by NMPA previously and one Class III solution was approved more recently in 2023 – even its real application is pretty limited by the description from NMPA. The statement is not completely true.
17. Line 417 – 422, it was said two-stage detector, while provide 3 point.
18. Please arrange the sequence of the supp tables and figures.
19. Please look at other remarks in the Methods, Figures, and Captions.
20. What is surface? Do you mean "interface"?

Reviewer #2 (Remarks to the Author):

The authors have answered all my comments appropriately. I believe the paper is suitable for publication now.

Reviewer #3 (Remarks to the Author):

In the revised version, my questions have been satisfactorily addressed, and appropriate modifications have been made. I have no further comments.

RESPONSE TO REVIEWERS' COMMENTS

Reviewer Comments and Detailed Response

Reviewer #1:

Comments to the Author

The revised manuscript appears to have undergone only minor amendments. The overall improvement in the quality of the manuscript is modest, and there are still many areas that need careful attention and revision. Consistency in terms, figures, tables, and clarity of presentation are crucial for readers' comprehension.

Abbreviation Consistency:

1. The abbreviation for the system name "Artificial Intelligence Cervical Cancer Screening" should be "AICCS" instead of "AICS." Please ensure the abbreviation is consistent throughout the manuscript.

Response: Thank you for your careful review of and helpful comment on our manuscript. We apologize for any confusion caused by the inconsistency of the system name. According to your suggestion, we have changed the abbreviation of the system name from "AICS" to "AICCS" throughout the revised manuscript.

2. On line 431, the system is referred to as "Automated Interpretation of Cervical Smears." If this is the correct full name, "AICS" is appropriate. Kindly verify and maintain consistency.

Response: Thank you for your detailed suggestions and kind reminder. We can't agree with you more that the abbreviation for the system name "Artificial Intelligence Cervical Cancer Screening" should be "AICCS" instead of "AICS", and we have revised throughout the manuscript and maintain consistency.

Paper Presentation:

1. Figures 1 and 2 are densely packed with information. I recommend labelling different sections for clarity and elaborating on the figure captions.

Response: Many thanks for your valuable comment. In response to your comment, we have revised Figure 1 and Figure2, and associated text to better elaborating the workflow of AICCS (Figure2, tracked version) and the algorithm of AICCS (Figure 3, tracked version). The revised Figure 1 (Figure2 in tracked version) is below for your reference.

(A) Annotation: The cervical liquid-based preparation smears were digitized, and digital pathology scanners generated WSIs. (B) Patch-level annotation: Cytopathologists then annotated abnormal cell grades, following the guidelines based on TBS 2014. These grades encompassed six main classes: ASC-US, LSIL, ASC-H, HSIL, SCC, and AGC. (C) WSI-level annotation: Patch-level cell detection results were utilized to detect features. Random forest classification was employed to generate a diagnosis based on the results obtained from the WSIs. Ultimately, a cervical cytology report was generated, summarizing the findings. AICCS, Artificial Intelligence Cervical Cancer Screening System. WSIs, whole slide images. ASC-US, atypical squamous cells of undetermined significance. LSIL, Low-grade squamous intraepithelial lesions. ASC-H, Atypical squamous cells-cannot exclude HSIL. HSIL, High-grade squamous intraepithelial lesions. SCC, Squamous cell carcinoma. AGC, Atypical glandular cells.

The revised Figure 2 (Figure3 in tracked version) is below for your reference.

(A) Patch level: Features were extracted from patch-level cell detection results. A random forest algorithm was used to produce a diagnosis through the classification of the WSI results. Entropy was chosen as the criterion for the model, for which the maximum depth was set to 10 and the maximum feature number was set to 50. The total estimate number was not allowed to exceed 200. (B) WSI level: The WSI-level classification model took the output results from the patch-

level cell detection model and produced one of five possible cytology grades, including NILM, ASC-US, LSIL, HSIL, and AGC. AICCS, Artificial Intelligence Cervical Cancer Screening System. WSI, whole slide image. NILM, Negative for intraepithelial lesion or malignancy. ASC-US, Atypical squamous cells of undetermined significance. LSIL, Low-grade squamous intraepithelial lesions. ASC-H, Atypical squamous cells-cannot exclude HSIL. HSIL, High-grade squamous intraepithelial lesions. SCC, Squamous cell carcinoma. AGC, Atypical glandular cells. SYSMH, Sun Yat-sen Memorial Hospital. GWCMC, Guangzhou Women and Children's Medical Center. TAHGMU, The Third Affiliated Hospital of Guangzhou Medical University.

2. The presentation of data in the results section needs revision. There is an unnecessary repetition of numerical values found in tables. It might be more effective to represent some of this data graphically. This lack of improvement was previously highlighted.

Response: We appreciate the reviewer's insightful comments on our manuscript. We acknowledge the importance of presenting some of the data graphically.

The specific modifications are as follows:

Methods and Materials section-- “The training dataset included 730 (7.8%) atypical squamous cells of undetermined significance (ASC-US), 995 (10.7%) low-grade squamous intraepithelial lesions (LSIL), 279 cases (3.0%) atypical squamous cells-cannot exclude a HSIL (ASC-H), 401 cases (4.3%) high-grade squamous intraepithelial lesions (HSIL), 35 (0.4%) squamous cell carcinoma (SCC), and 131 (1.4%) atypical glandular cells (AGC) cases (Figure 1A). In the internal validation dataset, 20.3% had intraepithelial lesions (Figure 1B). In external validation datasets, there are 44.7% and 32.7% had intraepithelial lesions in the GWCMC (Figure 1C) and TAHGMU datasets (Figure 1D), respectively (Table S1). Meanwhile, 4% of participants had intraepithelial lesions in SYSMH prospective validation datasets (Figure 1E, Table S1). In the randomized controlled trial with 608 participants, 2.80% had ASC-US, 2.0% had LSIL, 1.6% had HSIL, and 1.3% had AGC (Table S2), and the detailed proportions of intraepithelial lesions in AICCS alone, cytopathologist and AICCS-assisted group was shown in Figure 1F-Figure 1H.”. **(Page 6-7, line 111-122, tracked version)**

The Figure 1 (tracked version) is below for your reference.

Figure 1. Distribution of the cervical cytology grade in the training, validation datasets, and randomized controlled trial. NILM, Negative for intraepithelial lesion or malignancy. ASC-US, atypical squamous cells of undetermined significance. LSIL, Low-grade squamous intraepithelial lesions. ASC-H, Atypical squamous cells-cannot exclude HSIL. HSIL, High-grade squamous intraepithelial lesions. SCC, Squamous cell carcinoma. AGC, Atypical glandular cells.

3. Ensure that content is placed appropriately in the designated sections. I've noted specific concerns on the provided PDF.

Response: We appreciate the reviewer's careful comments on our manuscript. We have realized some content were placed inappropriately in the designated sections in the original version, and we modified accordingly.

The specific modifications are as follows:

Result section-- “For subgroup analysis, the category denoted as ASC-US+ encompasses ASC-US, LSIL, ASC-H, HSIL, and SCC. LSIL+ encompasses LSIL, ASC-H, HSIL, and SCC. Moreover, the HSIL category comprises ASC-H, HSIL, and SCC. For patients with Negative for intraepithelial lesion or malignancy (NILM) or with ASC-US+, the AICCS system achieved high AUC values (over 0.879) across all validation datasets (Figure 5A, Figure 5B). For patients with LSIL+, even higher AUCs were observed: 0.950 (0.933–0.967) on the internal dataset, 0.946 (0.924–0.968) on the GWCMC dataset, and 0.927 (0.896–0.959) on the TAHGMU dataset (Figure 5C). Among patients with HSIL, AUCs were 0.960 (0.938–0.982) on the internal dataset, 0.930 (0.899–0.962) on the GWCMC dataset, and 0.896 (0.816–0.975) on the TAHGMU dataset (Figure 5D).

For patients with LSIL+, the AICCS system’s sensitivity and accuracy were 0.928 (95% CI 0.894–0.954) and 0.883 (95% CI 0.868–0.896) on the internal dataset, 0.962 (95% CI 0.927–0.984) and 0.869 (95% CI 0.837–0.896) on the GWCMC dataset, and 0.955 (95% CI 0.904–0.983) and 0.845 (95% CI 0.812–0.874) on the TAHGMU dataset. Meanwhile, for patients with HSIL, sensitivity and accuracy were 0.980 (95% CI 0.942–0.996) and 0.882 (95% CI 0.867–0.897) on the internal dataset, 0.944 (95% CI 0.889–0.977) and 0.846 (95% CI 0.810–0.878) on the GWCMC dataset, and 0.889 (95% CI 0.708–0.977) and 0.815 (95% CI 0.776–0.850) on the TAHGMU dataset (Table 1).”. **(Page 8-9, line 149-165, tracked version)**

Result section-- “Among participants with LSIL+, the AICCS system and cytopathologists showed similar performance: a precision of 0.854 (95% CI 0.788–0.906) for the AICCS system and 0.868 (95% CI 0.804–0.918) for the cytopathologists (P=0.966). Similarly, for participants with HSIL, the AICCS system demonstrated a precision of 0.725 (95% CI 0.614–0.819), which was similar to that of the cytopathologists (0.740 (95% CI 0.626–0.834) (P= 0.899). Detailed results are presented in Table S6.”. **(Page 9, line 172-178, tracked version)**

REVIEWER COMMENTS

Reviewer #1 (Remarks to the Author):

Review comments

The authors have significantly improved the manuscript in this revision. The technical merits of the work are now more accessible and easier to evaluate. Additionally, the reorganization and enhancement of several figures have notably improved the manuscript's readability, while the Table 1, containing the key data, is still difficult to read and follow when reading the paper.

Paper Presentation:

1. Despite professional editing, the manuscript's language quality remains suboptimal and could benefit from further refinement. Authors are encouraged to consider additional improvements.
2. The data presented in Table 1, which contains crucial results, is challenging to interpret. Authors are advised to reconsider the presentation of this data in a more comprehensible format, as previously suggested. Effective data visualization is vital for a high-quality paper.
3. The current arrangement of figures and panels appears somewhat disorganized, leading to difficulty in following the paper's narrative. A more streamlined organization would enhance the reader's experience and prevent the need to frequently switch between figures and panels.
4. It would be beneficial to include a list of abbreviations at the beginning of the paper, rather than defining them repeatedly in figure captions. This approach would aid in reader comprehension.
5. Figure captions should be made self-explanatory, detailing the figures' significance without necessitating reference to the main text. Also, the organization of figures and panels in Figures 4-7 seems somewhat cluttered and would benefit from a more orderly presentation.

Major Concerns

1. Lines 124-130 state that the model was optimized, with certain training and model parameters detailed in Table S4. However, to justify the model selection and optimization, it is crucial to include the performance of different models in Table S4, which currently lacks such key information.
2. The manuscript lacks detailed information or a clear explanation regarding cell-level annotations. Specifically, in the section on cell detection (Line 429), models such as YOLO, SSD, and RetinaNet are mentioned. However, it remains unclear why RetinaNet with FPN was chosen as the backbone for the system's design, as there appears to be no supporting data provided for this decision. The verbal explanation of the advantages of RetinaNet is not sufficient to justify it is the optimal framework. Additionally, the results of the cell detection process are not presented. This omission is significant because understanding the outcomes of cell detection is crucial for evaluating the system's performance. Since cell detection is a fundamental step in whole slide image (WSI) classification, its results are essential to assess and comprehend the intermediate stages of the analysis.
3. Regarding the efficiency of AICCS mentioned in Line 218, it would be beneficial to include a quantitative comparison with traditional diagnostic speeds. Typically, cytologists take several minutes to examine a slide. Providing data to compare this average with the 120 seconds claimed for AICCS would strengthen the paper, as it highlights not only scientific conclusions but also the social and economic impacts.
4. The manuscript does not clearly address performance variations across three different sample preparation methods. It is important to specify whether AICCS is applicable to all three popular methods or if it is tailored to a specific one. If AICCS is developed for diverse methods, the

proportion of data from each and evidence of negligible performance differences should be presented. This is vital as some existing solutions are limited to one preparation method, with reduced effectiveness on others.

5. The text from Lines 310-315 raises concerns about the suitability of using thumbnail images for quality assessment. The practice among cytologists, as stated in Line 299, is to avoid evaluating slides with fewer than 5000 cells, suggesting that issues such as blurring and staining might be undetectable at very low resolutions. This brings into question the feasibility of accurately counting cell numbers in these low-resolution images. Therefore, it's essential to clarify the criteria used for quality control and provide details on the data used to train the EfficientNet for this task. The types of quality issues that can be discerned in low-resolution thumbnail images need to be specified. This aspect is particularly critical for the future, as any emerging industry will require a clearly defined QC standard, which is currently not well-established for AI in cytology diagnosis. More comprehensive information and data in this area are needed to significantly enhance the impact of this research. Additionally, it would be beneficial to consider making the QC solution/code publicly accessible, if feasible. Regarding WSI classification and the Random Forest solution, including a supplementary figure could offer a clearer presentation of these components. Please consider.

6. The manuscript should provide more details on colour augmentation, a key method for enhancing model performance. It's important to demonstrate how the chosen colour augmentation approach accounts for real-world variations, especially considering the potential significance of scanner variations compared to staining differences, if possible.

7. The manuscript lacks a description of the Whole Slide Image (WSI) classification process. Understanding that it employs a random forest-based method, this crucial aspect of the work should be explained and discussed in detail. I feel that the manuscript only described part of the work, if WSI level classification is explained.

8. The authors have made an effort to make some materials available, but this aspect could be further enhanced. Additionally, I suggest renaming the file in the provided link (<https://github.com/cellsvision/AICS>) to 'AICCS' instead of 'AICS' to maintain consistency with the system's name. The package has been downloaded and examined. It currently offers several networks and training code. However, I would recommend the authors also include testing code and some sample whole slide images (WSIs) for testing purposes. The absence of example images and other pertinent files makes it challenging to test and validate the solution described in the manuscript. Reproducibility is a crucial factor for a high-impact paper, particularly in the field of AI. Therefore, I would urge the authors to refer to Point 4 in the manuscript regarding the quality control (QC) method and consider including these details in the GitHub repository. Additionally, a better formatted and more comprehensive Readme file would greatly aid in understanding and using the provided materials.

Minor Issues

1. In Line 482, there is a discrepancy in the abbreviation of the system; it should be 'AICCS,' but the hyperlink incorrectly shows 'AICS.' Please correct this to ensure consistency throughout the document.
2. Regarding the author contributions in Line 554, there appears to be a discrepancy in the spelling of an author's name. It is listed as 'Heuri Yao,' but please verify if it should be 'Herui Yao.' It's crucial to check the entire paper to ensure that all authors' names are spelled correctly. Such low level mistakes should be prevented.
3. Line 328-329. I don't think it is appropriate to note the cytologist's name. Certainly I understand it is better to give the credits to the doctors who contributed to this work, while we also need to consider protect the clinicians and prevent any potential disputation in the future.
4. Line 149 and 154, it should be HSIL+ or HISL? Please confirm.

This is an important part of the work in the field of AI Pathology diagnosis, while there is still a margin for the authors to further improve the paper in a more systematic and detailed way. Please consider the above suggestions and comments. I am happy to have a look at a improved and

revised version again.

Some minor comments in the attached pdf files.

REVIEWER COMMENTS

Reviewer #1 (Remarks to the Author):

Review comments

The authors have significantly improved the manuscript in this revision. The technical merits of the work are now more accessible and easier to evaluate. Additionally, the reorganization and enhancement of several figures have notably improved the manuscript's readability, while the Table 1, containing the key data, is still difficult to read and follow when reading the paper.

Paper Presentation:

1. Despite professional editing, the manuscript's language quality remains suboptimal and could benefit from further refinement. Authors are encouraged to consider additional improvements.

Response: We sincerely appreciate your insightful suggestion, and sorry for the manuscript's language quality remains suboptimal. As a measure to address these concerns, the article has undergone an additional polish process by a professional agency, and we are pleased to provide the certificate of English language editing below for your reference.

2. The data presented in Table 1, which contains crucial results, is challenging to interpret. Authors are advised to reconsider the presentation of this data in a more comprehensible format, as previously suggested. Effective data visualization is vital for a high-quality paper.

Response: Thank you for your insightful suggestion, we do agree that Table 1 (in the previous manuscript) should be presented in a more comprehensible format which is crucial to improve the scientific level of this manuscript. Therefore, we visualized the data into Figure 5 (in the revised version) and made a clearer expression in the revised version (page 8, line 149-153): “The AICCS system demonstrated high sensitivity on all datasets. On the SYSMH dataset, its sensitivity was 0.906 (95% CI 0.875–0.932); on the GWCMC dataset, its sensitivity was 0.902 (95% CI 0.859–0.935); and on the TAHGMU dataset, its sensitivity was 0.918 (95% CI 0.868–0.953). Accuracy was also impressive: 0.881 (95% CI 0.866–0.894), 0.850 (95% CI 0.819–0.878), and 0.843 (95% CI 0.812–0.872) on the SYSMH, GWCMC, and TAHGMU datasets, respectively (Figure 5A). ” and page, line : “For patients with ASC-US+, the AICCS system’s sensitivity and accuracy were 0.911 (95% CI 0.879–0.936) and 0.881 (95% CI 0.867–0.895) on the internal dataset, 0.935 (95% CI 0.927–0.984) and 0.861 (95% CI 0.829–0.888) on the GWCMC dataset, and 0.925 (95% CI 0.876–0.960) and 0.844

(95% CI 0.813–0.873) on the TAHGMU dataset (Figure 5B). For patients with LSIL+, the AICCS system’s sensitivity and accuracy were 0.928 (95% CI 0.894–0.954) and 0.883 (95% CI 0.868–0.896) on the internal dataset, 0.962 (95% CI 0.927–0.984) and 0.869 (95% CI 0.837–0.896) on the GWCMC dataset, and 0.955 (95% CI 0.904–0.983) and 0.845 (95% CI 0.812–0.874) on the TAHGMU dataset (Figure 5C). Meanwhile, for patients with HSIL+, sensitivity and accuracy were 0.980 (95% CI 0.942–0.996) and 0.882 (95% CI 0.867–0.897) on the internal dataset, 0.944 (95% CI 0.889–0.977) and 0.846 (95% CI 0.810–0.878) on the GWCMC dataset, and 0.889 (95% CI 0.708–0.977) and 0.815 (95% CI 0.776–0.850) on the TAHGMU dataset (Figure 5D)”. The Figure 5 below is for your reference.

3. The current arrangement of figures and panels appears somewhat disorganized, leading to difficulty in following the paper's narrative. A more streamlined organization would enhance the reader's experience and prevent the need to frequently

switch between figures and panels.

Response: We appreciate the reviewer's careful comments on our manuscript. We have realized that the current arrangement of figures and panels were disorganized which was a challenge for understanding. Therefore, we have rearranged the figures and tables accordingly.

4. It would be beneficial to include a list of abbreviations at the beginning of the paper, rather than defining them repeatedly in figure captions. This approach would aid in reader comprehension.

Response: Many thanks for your valuable comment. In response to your comment, we have added a Table S3 in the beginning of the paper including a list of abbreviations. The Table S3 below is for your reference.

Table S3. Abbreviation and definitions.

Abbreviation	Definition
ADC	Adenocarcinoma
AGC	Atypical glandular cells
AGC-FN	Atypical glandular cells, favor neoplastic
AGC-NOS	Atypical glandular cells, not otherwise specified
AICCS	Artificial Intelligence Cervical Cancer Screening
AIS	Endocervical adenocarcinoma in situ
ASC-H	Atypical squamous cells - cannot exclude a HSIL
ASC-US	Atypical squamous cells of undetermined significance
AUC	Area under the curve
CNNs	Convolutional Neural Networks
FPN	Feature Pyramid Network
GWCMC	Guangzhou Women and Children Medical Center
HDFP	Hybrid Deep Feature Fusion
HSIL	High-grade squamous intraepithelial lesions
LSIL	Low-grade squamous intraepithelial lesions
NILM	Negative for intraepithelial lesion or malignancy
NMPA	The National Medical Products Administration
NPV	Negative predictive value
POC	Proof-of-concept
ROIs	Regions of interest

SaaS	Software-as-a-Service
SCC	Squamous cell carcinoma
SDA	Stain density absorbance
SYSMH	Sun Yat-sen Memorial Hospital
TAHGMU	The Third Affiliated Hospital of Guangzhou Medical University
TBS	The Bethesda System
WSI	Whole-slide image

5. Figure captions should be made self-explanatory, detailing the figures' significance without necessitating reference to the main text. Also, the organization of figures and panels in Figures 4-7 seems somewhat cluttered and would benefit from a more orderly presentation.

Response: Thank you for your careful review of and helpful comment on our manuscript. We apologize for the blurry figure captions and disorganized figures and panels in the Figure 4-7 in the previous version. In response to your suggestions, the figures were rearranged and captions were modified. The Figure 4 in the the previous version was deleted for its not benefit for streamline presentation. And the update version of Figure 4-7 were as follow:

Figure 4. ROC curves of AICCS system according to risk stratification obtained on different validation datasets.

ROC curves obtained by the AICCS system for participants with (A) ASC-US+, (B) LSIL+, and (C) HSIL+. ROC, Receiver operating characteristic curve. AICCS, Artificial Intelligence Cervical Cancer Screening System. ASC-US, Atypical squamous cells of undetermined significance. LSIL, Low-grade squamous intraepithelial lesions. HSIL, High-grade squamous intraepithelial lesions. ASC-US+ includes ASC-US, LSIL, ASC-H, HSIL, and SCC. LSIL+ includes LSIL, ASC-H, HSIL, and SCC. HSIL+ includes ASC-H, HSIL and SCC. SYSMH, Sun Yat-sen Memorial Hospital. GWCMC, Guangzhou Women and Children's Medical Center. TAHGMU, the Third Affiliated Hospital of Guangzhou Medical University.

Figure 5. Overall performance on retrospective validation datasets.

(A) Overall performance of the AICCS system on all cytology grades among different validation datasets. (B) Overall performance of the AICCS system on ASC-US+ among different validation datasets. (C) Overall performance of the AICCS system on LSIL+ among different validation datasets. (D) Overall performance of the AICCS system on HSIL+ among different validation datasets. AICCS, Artificial Intelligence Cervical Cancer Screening System. ASC-US, Atypical squamous cells of

undetermined significance. LSIL, Low-grade squamous intraepithelial lesions. HSIL, High-grade squamous intraepithelial lesions. ASC-US+ includes ASC-US, LSIL, ASC-H, HSIL, and SCC. LSIL+ includes LSIL, ASC-H, HSIL, and SCC. HSIL+ includes ASC-H, HSIL and SCC. SYSMH, Sun Yat-sen Memorial Hospital. GWCMC, Guangzhou Women and Children’s Medical Center. TAHGMU, the Third Affiliated Hospital of Guangzhou Medical University.

Figure 6. Comparisons of diagnostic performance of the AICCS alone, cytopathologists, and AICCS-assisted cytopathologists on the prospective validation dataset.

Participants with (A) ASC-US+, (B) LSIL+, and (C) HSIL+. AICCS, Artificial intelligence cervical cancer screening System. ASC-US, Atypical squamous cells of undetermined significance. LSIL, Low-grade squamous intraepithelial lesions. HSIL, High-grade squamous intraepithelial lesions. ASC-US+ includes ASC-US, LSIL, ASC-H, HSIL, and SCC. LSIL+ includes LSIL, ASC-H, HSIL, and SCC. HSIL+ includes ASC-H, HSIL and SCC.

A. Performance of AICCS and cytopathologist in ASC-US+

B. Performance of AICCS and cytopathologist in LSIL+

C. Performance of AICCS and cytopathologist in HSIL+

Figure 7. Comparisons of diagnostic performance of the AICCS alone, cytopathologists, and AICCS-assisted cytopathologists in the randomized controlled trial.

Participants with (A) ASC-US+, (B) LSIL+, and (C) HSIL+. AICCS, Artificial intelligence cervical cancer screening System. ASC-US, Atypical squamous cells of undetermined significance. LSIL, Low-grade squamous intraepithelial lesions. HSIL, High-grade squamous intraepithelial lesions. ASC-US+ includes ASC-US, LSIL, ASC-H, HSIL, and SCC. LSIL+ includes LSIL, ASC-H, HSIL, and SCC. HSIL+ includes ASC-H, HSIL and SCC.

A. Performance of AICCS and cytopathologist in ASC-US+

B. Performance of AICCS and cytopathologist in LSIL+

C. Performance of AICCS and cytopathologist in HSIL+

Major Concerns

1. Lines 124-130 state that the model was optimized, with certain training and model parameters detailed in Table S4. However, to justify the model selection and optimization, it is crucial to include the performance of different models in Table S4, which currently lacks such key information.

Response: Many thanks for your valuable comment. We couldn't agree more that the performance of different models is crucial for justify the model selection and optimization, therefore the related content was reported in Table S3 in the Supplement of the previous manuscript. As include the performance of different models with the model parameters will be more comprehensive in understanding the model selection and optimization. Therefore, we have revised the table combing the model performance and model parameters to better elaborating the details of model selection and optimization, and the revised Table S5 is below for your reference.

Table S5. Performance of four deep learning algorithms in cervical cytopathological diagnosis

	Cell detection			WSI classification			Sensitivity (95% CI)	Specificity (95% CI)	Accuracy (95% CI)	NPV (95% CI)	PPV (95% CI)	AUC (95% CI)
	Network	Backbone	Hyperparameters	Features	Classifier	Hyperparameters						
Model 1	Faster-RCNN	ResNet50	Patch size: 1,024×1,024 Lr: 0.0001–0.00001 Optimizer: sgd Momentu: 0.90	Cell distributions	DNN	Layer: 383×512×512×5 Lr: 0.0001 Optimizer: sgd Momentu:0.90 Drop rate: 0 Cv:5 Class_weight: {0: 0.3, 1: 1, 2: 0.8, 3: 0.9, 4: 2} criterion: entropy Max_depth: 30 Max_features: 80 N_estimators: 300	0.711 (0.665–0.754)	0.797 (0.777–0.816)	0.780 (0.762–0.798)	0.919 (0.904–0.932)	0.460 (0.420–0.499)	0.830 (0.805–0.855)
Model 2	Faster-RCNN	ResNet50	Patch size: 1,024×1,024 Lr: 0.0001–0.00001 Optimizer: sgd Momentu: 0.90	Cell distributions	Random forest	Layer: 383×256×256×5 Lr: 0.0001 Optimizer: sgd Momentu:0.90 Drop rate: 0.20 Cv: 4 Class_weight: {0: 0.2, 1: 0.5, 2: 1, 3: 1, 4: 0.8}	0.836 (0.796–0.871)	0.889 (0.873–0.903)	0.879 (0.864–0.892)	0.959 (0.948–0.968)	0.636 (0.593–0.677)	0.918 (0.899–0.937)
Model 3	Retina	ResNet18	Patch size: 1,024×1,024 Lr: 0.01–0.002 Optimizer: sgd Momentu: 0.90	Cell distributions	DNN	Layer: 383×256×256×5 Lr: 0.0001 Optimizer: sgd Momentu:0.90 Drop rate: 0.20 Cv: 4 Class_weight: {0: 0.2, 1: 0.5, 2: 1, 3: 1, 4: 0.8}	0.538 (0.490–0.585)	0.945 (0.933–0.955)	0.862 (0.847–0.876)	0.889 (0.874–0.903)	0.712 (0.660–0.760)	0.830 (0.805–0.854)
Model 4	Retina	ResNet18	Patch size: 1,024×1,024 Lr: 0.01–0.002 Optimizer: sgd	Cell distributions	Random forest	Layer: 383×256×256×5 Lr: 0.0001 Optimizer: sgd Momentu:0.90 Drop rate: 0.20 Cv: 4 Class_weight: {0: 0.2, 1: 0.5, 2: 1, 3: 1, 4: 0.8}	0.906 (0.875–0.932)	0.874 (0.857–0.889)	0.881 (0.866–0.894)	0.980 (0.971–0.987)	0.647 (0.608–0.685)	0.922 (0.904–0.940)